# Nanoparticles as an antidote for poisoned gold single-atom catalysts in sustainable propylene epoxidation

Qianhong Wang[1,3], Keng Sang[1,3], Changwei Liu[1,3], Zhihua Zhang[1], Wenyao Chen [1] ✉, Te Ji[2], Lina Li[2], Cheng Lian [1], Gang Qian[1], Jing Zhang[1], Xinggui Zhou [1] ✉, Weikang Yuan[1] & Xuezhi Duan [1] ✉

The development of sustainable and anti-poisoning single-atom catalysts (SACs) is essential for advancing their research from laboratory to industry. Here, we present a proof-of-concept study on the poisoning of Au SACs, and the antidote of Au nanoparticles (NPs), with trace addition shown to reinforce and sustain propylene epoxidation. Multiple characterizations, kinetics investigations, and multiscale simulations reveal that Au SACs display remarkable epoxidation activity at a low propylene coverage, but become poisoned at higher coverages. Interestingly, Au NPs can synergistically cooperate with Au SACs by providing distinct active sites required for $H_2/O_2$ and $C_3H_6$ activations, as well as hydroperoxyl radical to restore poisoned SACs. The difference in reaction order between $C_3H_6$ and $H_2$ ($n_{C3H6}$-$n_{H2}$) is identified as the descriptor for establishing the volcano curves, which can be fine-tuned by the intimacy and composition of SACs and NPs to achieve a rate-matching scenario for the formation, transfer, and consumption of hydroperoxyl. Consequently, only trace addition of Au NPs antidote (0.3% ratio of SACs) stimulates significant improvements in propylene oxide formation rate, selectivity, and $H_2$ efficiency compared to SACs alone, offering a 56-fold, 3-fold, and 22-fold increase, respectively, whose performances can be maintained for 150 h.

Single-atom catalysts (SACs) capture significant interest in catalysis by virtue of their maximum atom utilization, tunable electronic properties, and special size quantum effects, which afford superior catalytic performances across a wide range of catalytic reactions[1–4]. Despite various approaches developed for the large-scale synthesis of SACs with uniform and well-defined atomic structures, SACs are not guaranteed to provide a more efficient and sustainable catalytic conversion than their nanoparticle counterparts[5–7]. Apart from their poor thermal stability under severe conditions, SACs encounter notable obstacles including variable reactant adsorption capabilities and limited active sites[8]. On the one hand, SACs may exhibit either too strong or too weak adsorption capability for certain reactants or intermediates[9–11]. On the other hand, they fail to provide the cooperativity between different types of active sites required for complex reactions that involve the activation, adsorption and migration of reactants and intermediates[12–14]. These two difficulties pose significant challenges for SACs in reactions involving multiple reactants or the formation of intermediates that require stabilization before undergoing subsequent reactions[15,16]. Hence, it is of fundamental importance to explore and develop new strategies to overcome these two difficulties for extending the SACs from laboratorial synthesis to industrial application.

[1]State Key Laboratory of Chemical Engineering, East China University of Science and Technology, 130 Meilong Road, Shanghai 200237, China. [2]Shanghai Synchrotron Radiation Facility, Shanghai Advanced Research Institute, Shanghai 201210, China. [3]These authors contributed equally: Qianhong Wang, Keng Sang, Changwei Liu. ✉e-mail: wenyao.chen@ecust.edu.cn; xgzhou@ecust.edu.cn; xzduan@ecust.edu.cn

The direct epoxidation of propylene with $H_2/O_2$ to value-added propylene oxide (PO) using bifunctional Au-Ti catalysts ($C_3H_6 + H_2 + O_2 \rightarrow C_3H_6O + H_2O$) is considered a dream reaction for PO production[17,18]. It is widely accepted that hydrogen peroxide or hydroperoxyl intermediate generated by $H_2$ and $O_2$ on Au sites would transfer and react with $C_3H_6$ on adjacent Ti sites to generate PO, and neither Au nor Ti alone would give any significant PO activity[19,20]. As such, the manipulation of Ti coordination environment and Au particle size are the two main tools to improve catalytic performances[21–24]. However, there is currently an intense debate on the role of Au particle size in this reaction, as different sizes ranging from 2–5 nm[25], 1–2 nm[26], 1.4 nm[27], ~1 nm[28], <1 nm[29], or 0.7 nm[30] have been suggested as Au active site. Rationally, this debate can be attributed to the use of different Ti supports ($TiO_2$, TS-1, Ti-SBA-15, $TiO_2/SiO_2$, and Ti-$SiO_2$) and related preparation details (Si/Ti ratio, calcination, and pore structure), yielding different Au-Ti synergies and catalytic results. Moreover, as far as we are concerned, tiny Au clusters, even Au single atoms (SAs), have rarely been tested for this reaction, despite $Au_3$ being predicted as an active site based on DFT calculations[31]. Hence, it is highly desirable to eliminate the Ti effects to obtain a thorough understanding of Au size effects from nanoscale to subnanoscale and even single atom, which will help form a complete picture to refine the design principle of Au catalyst.

In this study, we report the fabrication of a series of Au catalysts anchored on Ti-free silicalite-1 (S-1) support to obtain the isolated Au SAs ($Au_1$), Au NPs ($Au_n$), as well as their coexistences ($Au_{1\&n}$) for propylene epoxidation. Both the PO formation rate and selectivity, as well as $H_2$ efficiency exhibit volcano curves as a function of Au size, with $Au_{1\&n}$ exhibiting the best catalytic performance. It is demonstrated that Au SACs display an attractive epoxidation activity at a low propylene surface coverage, but become poisoned by its strong adsorption at higher coverage. However, Au NPs can provide hydroperoxyl radical for reacting with $C_3H_6$ on Au SAs, thus lowering $C_3H_6$ coverage to restore Au SAs. Simultaneously, Au NPs can cooperate with Au SAs to provide different types of active sites required for the activation of $H_2/O_2$ and $C_3H_6$, respectively. Hence, $Au_n$ can serve as the antidote for $Au_1$ catalyst to overcome the above two representative problems associated with SACs. Furthermore, the difference in reaction order between $C_3H_6$ to $H_2$ ($n_{C_3H_6}$-$n_{H_2}$) is identified as catalytic descriptor to establish the volcano curves, which can be fine-tuned by the intimacy and composition of Au SACs and Au NPs to achieve the reactant coverage-matching. As a result, only trace addition of Au NPs (0.3% ratio of Au SACs) can stimulate a simultaneous increase in PO formation rate from 10.4 to 583.6 $mol_{PO}\cdot mol_{Au}^{-1}\cdot h^{-1}$, PO selectivity from 25.1%–75.6%, and $H_2$ efficiency from 1.4% to 30.5%, whose catalytic performances can be maintained for 150 h.

## Results

### Catalyst preparation and characterization

As discussed above, the prerequisite to make a fair comparison of Au size effects remains to alleviate the disturbance of Ti species. However, this contradicts the formation of PO over Au-Ti bifunctional catalysts, while the monometallic Au in the absence of Ti primarily yields acrolein[32,33]. This discrepancy was first explored by first-principles theoretical calculations with periodic density functional theory (DFT) on a representative Au(111) surface. We calculated a variety of possible adsorption sites, and identified the reaction pathways starting from propylene and hydroperoxyl based on previous studies[34,35]. As a result, two pathways for PO formation were considered in Supplementary Fig. 1, involving hydroperoxametallacycle (HO-OMC) formation or hydroperoxyl dissociation, both leading to the same precursor of a five-member ring oxametallacycle (OMMC) and a subsequent four-member one (OMC). The transformation of OMMC into PO was identified as the rate-determining step (RDS) with an energy barrier of 1.11 eV. For acrolein formation (Supplementary Fig. 2), the RDS involves

the attack of allyl intermediates by oxygen from hydroperoxyl dissociation with the energy barrier of 1.10 eV. The similar energy barrier for PO and acrolein generation indicates that PO has the similar possibility as acrolein to be produced on monometallic Au, which motivates us to study Au size effects in the absence of Ti.

Accordingly, using uncalcined S-1 as support to suppress catalyst deactivation from pore blocking by carbonaceous deposits, we prepared a series of Au/S-1 catalysts by deposition-precipitation method, and the nominal and actual Au loadings were summarized in Supplementary Table 1. As characterized by HAADF-STEM in Supplementary Figs. 3–10, lowering Au loadings from 0.460 wt% to 0.026 wt% can decrease Au particle size from 3.9 to 2.1 nm (Supplementary Table 1), based on the measurement of >150 random particles. The high dispersion and small size of Au NPs were also confirmed by the XRD patterns in Supplementary Fig. 11. As shown in Fig. 1a–c, further decreasing Au loading from 0.014 wt% to 0.004 wt% significantly lowers Au NPs density and size, making it difficult to distinguish Au sites. Hence, aberration-corrected HAADF-STEM was employed to characterize the catalysts with the loading of 0.004, 0.014, and 0.026 wt%, which consist of isolated Au SAs (labeled by the red circle in Fig. 1d and Supplementary Fig. 10), a mixture of Au SAs and NPs (Fig. 1e), and Au NPs (labeled by the blue square in Fig. 1f), respectively. In this regard, the three catalysts were denoted as $Au_1$, $Au_{1\&n}$, and $Au_n$, in which the absence of Au NPs for $Au_1$ catalyst is consistent with its ultralow loading (0.004 wt%).

The XPS spectra in Fig. 1g exhibit a continuous increase of Au 4$f$ binding energy by decreasing Au size, most likely attributed to the final state effect resulting from the size-dependent electrostatic interaction between ionized cluster and escaping photoelectron[36]. Moreover, the negative shift of Au 4$f$ binding energy of these catalysts compared with bulk Au indicates electron transfer from S-1 – Au nanoparticles for these catalysts, and the resultant electron-rich Au species has been suggested to weaken oxygen adsorption to promote this reaction[37]. Correspondingly, there is a significant decrease in signal intensity, particularly for $Au_1$, whose signal was too weak to be detectable. Notably, although there are some isolated Au SAs in the $Au_{1\&n}$ catalyst, most of Au atoms are ensembled, and the metallic character still dominates, similar to the $Au_n$ catalyst. Unfortunately, XAS failed to gain more insights into their electronic structures and coordination environments due to the low Au loadings. In this regard, we conducted in-situ synchrotron-based FTIR (SR-FTIR) measurements with a smaller spot size and faster acquisition capability, enabling us to obtain high-quality spectral imaging data from a synchrotron light source. A comparison of the C−H stretching vibrations of $C_3H_6$, represented by the band ranging from 2935 – 3015 $cm^{-1}$ [38], was made in Fig. 1h. $Au_1$ only exhibits a slightly decreased adsorption of $C_3H_6$ compared to $Au_n$, despite its ultralow Au loading. However, Fig. 1i reveals a noticeable red shift of 2 $cm^{-1}$ for the $Au_n$ catalyst with respect to the $Au_1$ catalyst. This shift is most likely attributed to the higher electron density of Au NPs compared to Au SAs, agreeing well with the above Au 4$f$ binding energy shift.

### Catalytic performances and kinetics investigations

The catalytic performances of these catalysts for propylene epoxidation were summarized in Supplementary Table 2, and compared in Fig. 2a. Both PO formation rate and selectivity as well as $H_2$ efficiency exhibit volcano curves as a function of Au loading, with $Au_{1\&n}$ catalyst providing the highest catalytic performance of 80.2 $mol_{PO}\cdot mol_{Au}^{-1}\cdot h^{-1}$ than $Au_1$ (10.4 $mol_{PO}\cdot mol_{Au}^{-1}\cdot h^{-1}$) and $Au_n$ (25.9 $mol_{PO}\cdot mol_{Au}^{-1}\cdot h^{-1}$) catalyst. Moreover, PO emerges as the main product for $Au_{1\&n}$ catalyst with the selectivity of 55.2%, while propanal and acrolein dominate the product distribution at the left and right side of the volcano curve, respectively. After minimizing the effects of internal and external diffusion limitations as shown in Supplementary Table 3, the activation energy for each catalyst was determined based on the corresponding

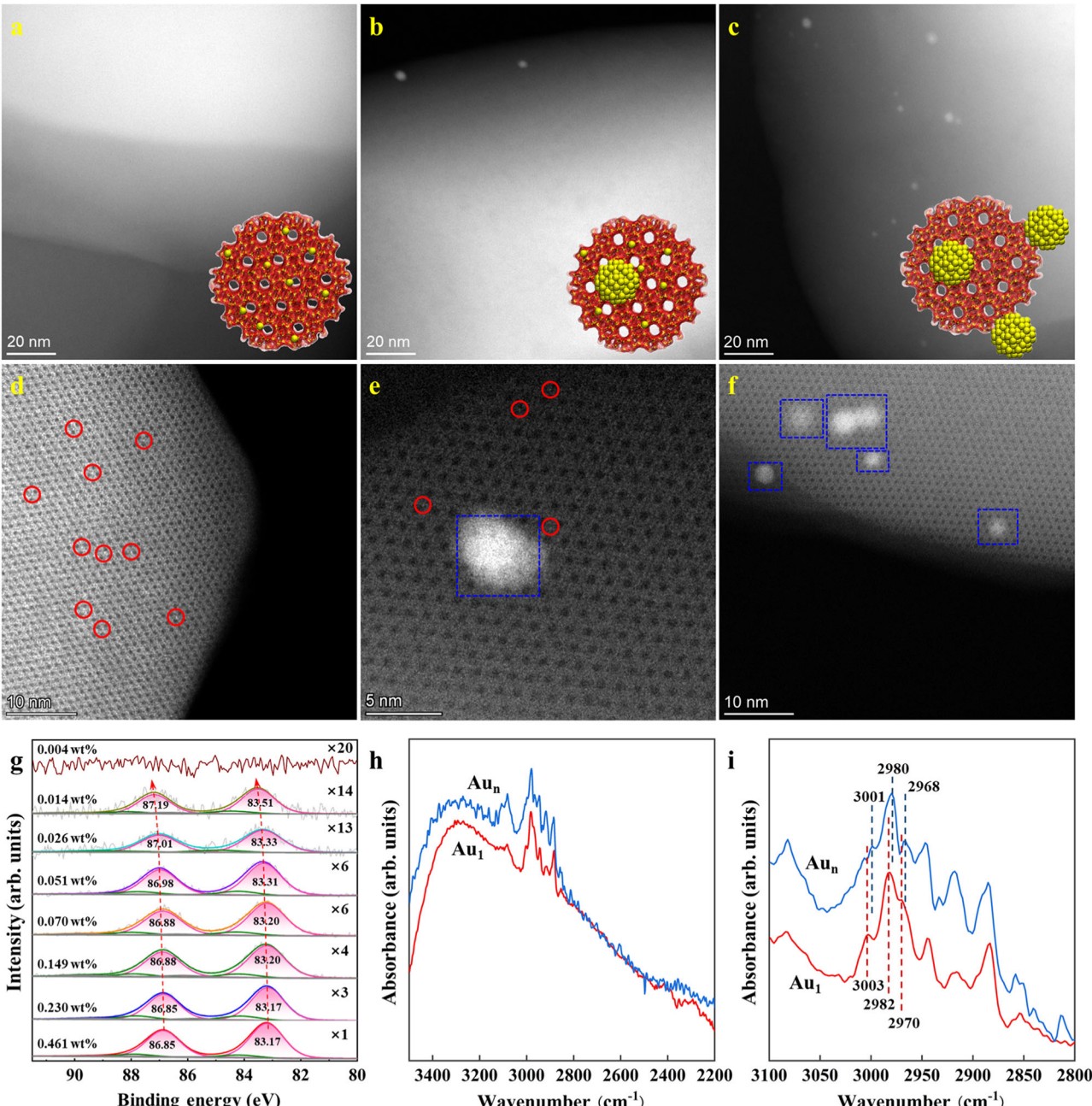

**Fig. 1 | Structural characterization of Au₁, Au₁&ₙ, and Auₙ. a–c** HAADF-STEM images in low magnification of $Au_1$ (**a**), $Au_{1\&n}$ (**b**), and $Au_n$ (**c**) catalyst. **d–f** Aberration-corrected HAADF-STEM images in high magnification of $Au_1$ (**d**), $Au_{1\&n}$ (**e**), and $Au_n$ (**f**) catalyst. **g**, XPS Au 4$f$ spectra of Au/S-1 catalysts with different loadings. **h, i** In-situ SR-FTIR spectra of $C_3H_6$ adsorption (**h**) and that in the range of 2800-3100 cm⁻¹ (**i**) of the $Au_1$ and $Au_n$ catalyst at 40 °C. The Au loadings for the $Au_1$, $Au_{1\&n}$, and $Au_n$ catalysts are 0.004, 0.014, and 0.026 wt%, respectively.

Arrhenius plots in Supplementary Fig. 12, and is shown in Fig. 2b, in which $Au_{1\&n}$ catalyst has the lowest activation energy (23.0 kJ·mol⁻¹) followed by $Au_n$ (28.7 kJ·mol⁻¹) and $Au_1$ (46.3 kJ·mol⁻¹). Interestingly, the activation energy further decreases with increasing Au loading, reaching a minimal of 9.4 kJ·mol⁻¹ for the catalyst with the loading of 0.460 wt%. The decrease in activation energy seems in contrast to the lowest catalytic performance observed for this catalyst, suggesting that reactant adsorption, rather than activation, can be the main cause for its poor catalytic performance in Fig. 2a. Hence, in addition to studying the effects of reaction temperature in related to the activation, we have investigated the effects of reactants concentration in related to the adsorption as shown in Supplementary Figs. 13–15, and the resultant reaction orders are further compared in Fig. 2c.

Interestingly, the reaction order of either $H_2$ or $C_3H_6$ varies dramatically with Au loading, where that of $H_2$ continuously decreases from 0.73 to -0.40, compensated by an increase of $C_3H_6$ from 0.20 to 0.78. Comparatively, the reaction order for $O_2$ exhibits a slight decrease regardless of the amount of Au loaded onto the catalyst. All of these observations indicate a negligible change in $O_2$ surface coverage, but a strong competitive adsorption between $H_2$ and $C_3H_6$ on the catalyst surface. Specifically, the surfaces of $Au_1$ and $Au_n$ catalyst are mainly covered by $C_3H_6$ and $H_2$, respectively, consistent with the above in-situ SR-FTIR results. Moreover, an unexpected shift in kinetic isotope effects (KIE) from normal ($Au_1$ catalyst) to inverse ones ($Au_n$ catalyst) is observed in Fig. 2d. In general, inverse KIE are typically associated with variations in the steric environment of the active site

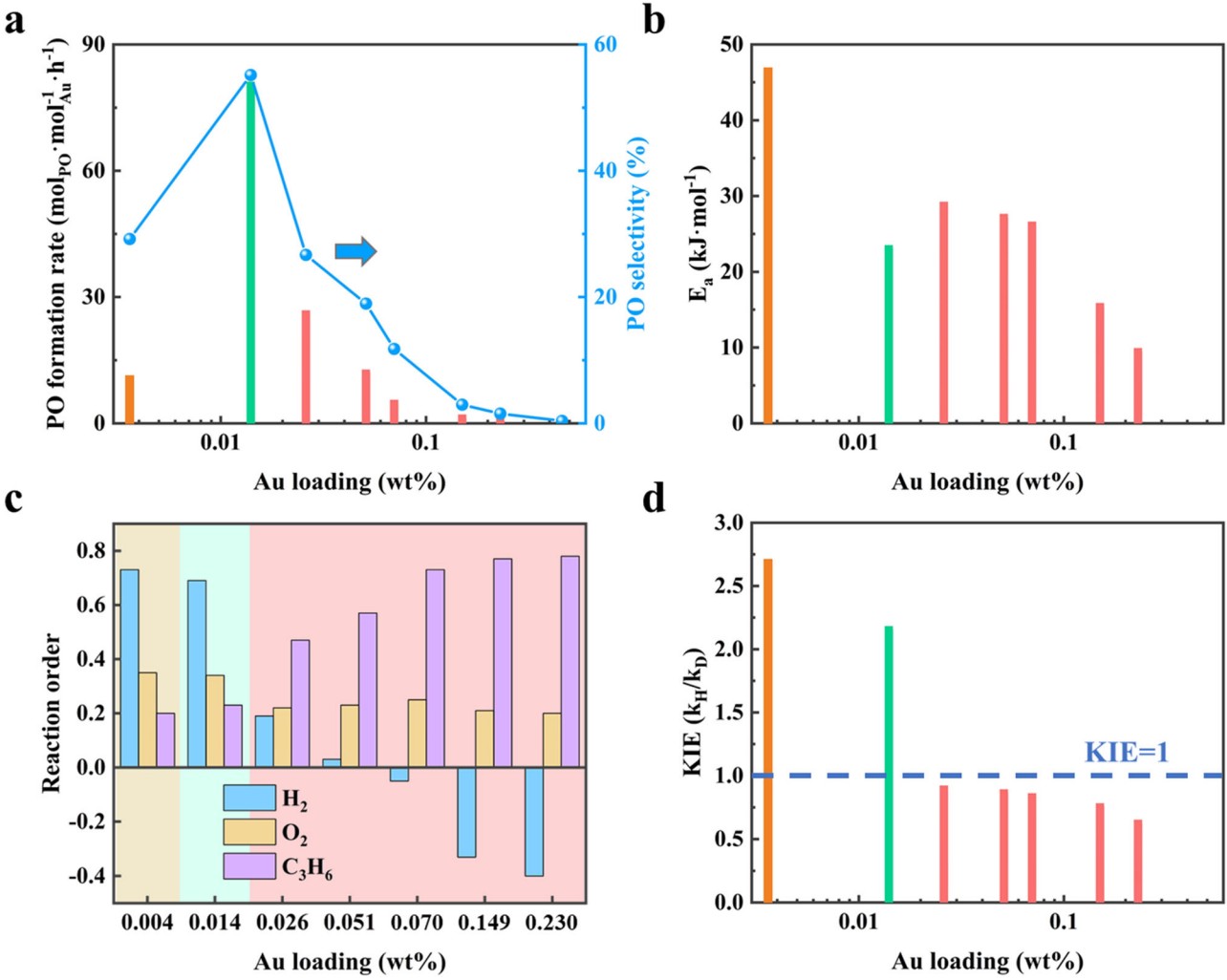

**Fig. 2 | Catalytic performance and kinetic investigations. a–d** PO formation rate and selectivity (**a**), activation energy (**b**), reaction order of $H_2$, $O_2$, and $C_3H_6$ (**c**), as well as kinetic isotope effects (**d**) of the Au/S–1 catalysts with different loadings.

due to deuteration during RDS[39–41]. In this case, since the D atom is heavier than the H atom, the stronger D-D and C-D bonds could result in shorter bond lengths compared to the H-H and C-H bonds[42]. Consequently, the shorter D-D and C-D bonds can lead to a less obstructed active site and facilitates a less hindered pathway for reactants, resulting in the observed inverse KIE for the $Au_n$ catalyst, which aligns with its surface being covered by hydrogen. Accordingly, the $Au_1$ and $Au_n$ catalysts under reaction conditions are mainly covered by $C_3H_6$ and $H_2$, respectively. This explains why the Au catalyst with the highest loading (0.460 wt%) is probably poisoned by $H_2$ to exhibit the most negative reaction order of -0.40 as well as the poorest activity, despite having the lowest activation energy. In contrast, the moderate reaction orders of $C_3H_6$ and $H_2$, as well as KIE, suggest a trade-off of surface coverage between $C_3H_6$ and $H_2$ for the $Au_{1\&n}$ catalyst, which affords its highest catalytic performance among these catalysts.

## Mechanistic insights into the poisoning of Au SACs

To gain mechanistic insights into the significantly different epoxidation activity among the $Au_1$, $Au_{1\&n}$, and $Au_n$ catalyst, we first performed DFT calculations on Au single atom, $Au_{13}$ cluster, and Au(111) surface. Accordingly, Au single atom and $Au_{13}$ cluster models on silicalite-1 surface were constructed via an investigation of different local structures of Au species in Supplementary Figs. 16, 17, respectively. The charge population for Au atom, obtained by integrating the projected density of state (PDOS) up to the Fermi level as shown in

Supplementary Fig. 18, is determined to be 8.82 e. This implies an almost complete transfer of one *d* electron from the Au atom to S-1, which is consistent with the observed binding energy shift in XPS spectra. The reaction pathways of propylene epoxidation on Au single atom, $Au_{13}$ cluster, and Au(111) surface are exhibited in Fig. 3a, with the corresponding structural configurations displayed in Supplementary Figs. 19–21. It can be seen that all these models initiate with the reaction between $O_2$ and $H_2$ to generate hydroperoxyl radical and hydrogen adatom. Subsequently, the hydroperoxyl dissociates into the active oxygen species over Au(111), which will react with $C_3H_6$ to form oxametallacycle as propylene oxide precursor. The rate-determining step (RDS) is identified as the hydroperoxyl formation ($O_2^* + H_2^* \rightarrow OOH^* + H^*$) with an energy barrier of 1.57 eV. In contrast, the active oxygen species derived from hydroperoxyl dissociation can directly react with $C_3H_6$ to produce PO over $Au_{13}$ cluster without the formation of oxametallacycle, which is identified as the RDS with an energy barrier of 0.25 eV. Interestingly, the Au single atom can further simply the reaction pathway by directly catalyzing the reaction between the hydroperoxyl radical and $C_3H_6$ into PO with the lowest energy barrier (0.20 eV), shifting the RDS back to the hydroperoxyl formation with an energy barrier of 0.31 eV.

As shown in Fig. 3b, the smaller-sized Au species including Au single atom and $Au_{13}$ cluster can greatly lower the energy barrier to promote PO generation. This is consistent with the higher activity of subnano-sized metal species compared to metal NPs in many

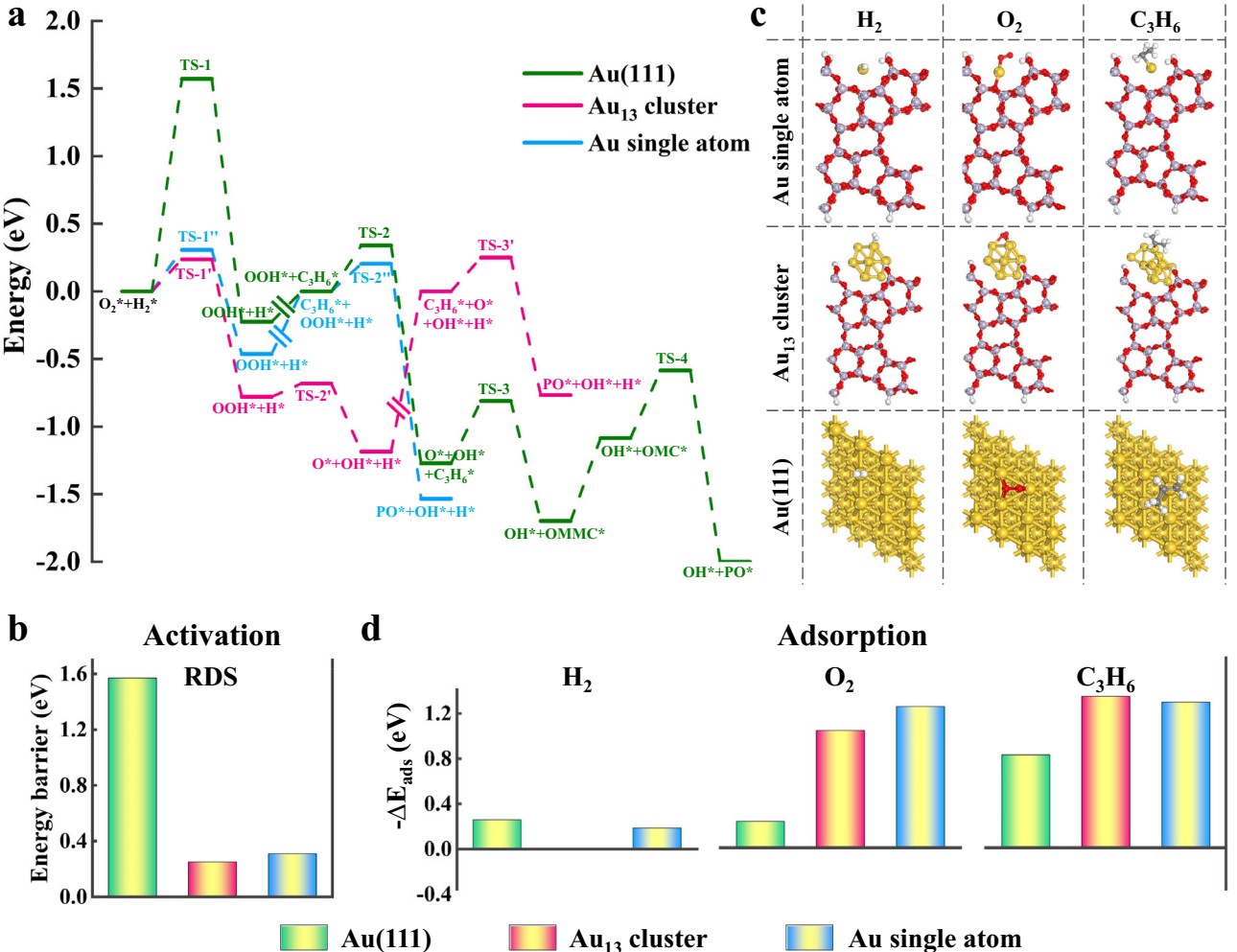

**Fig. 3 | Theoretical calculation on propylene adsorption and activation.**
**a**–**d** Calculated energy profile for propylene epoxidation (**a**), the corresponding energy barrier of rate-determining step (**b**), most energy-favorable adsorption configurations of $O_2$, $H_2$, and $C_3H_6$ (**c**), and the corresponding adsorption energies ($\Delta E_{ads}$) (**d**) over Au single atom, $Au_{13}$ cluster, and Au(111) surface.

reactions. However, this contradicts the poorest catalytic performance of $Au_1$ catalyst with the highest composition of Au SAs than $Au_{1\&n}$ and $Au_n$ catalyst. According to the kinetics investigation described above, not only the activations of reaction species but also their adsorptions have profound effects on this reaction. In this regard, we further compared the adsorption of reactants and products over Au single atom, $Au_{13}$ cluster, and Au(111) surface. The adsorptions of $H_2$, $O_2$, and $C_3H_6$ were investigated in Supplementary Figs. 22–24, and their adsorption geometries were computed for the top, bridge, and 3-fold hollow sites. Accordingly, the most energy-favorable adsorption configurations are displayed in Fig. 3c, and the corresponding adsorption energies are summarized in Fig. 3d. Obviously, $H_2$ binds very weakly with Au single atom, $Au_{13}$ cluster, and Au(111) surface. In comparison, both $O_2$ and $C_3H_6$ exhibit much higher adsorption energies on Au single atom, $Au_{13}$ cluster than them on Au(111) surface. Accordingly, Au single atom prefers to strongly interact with $C_3H_6$ rather than $H_2$, as also suggested by the low reaction order for $C_3H_6$ and high reaction order for $H_2$.

Hence, based on DFT calculations, it is suggested that Au SAs exhibit high catalytic activity for propylene epoxidation at low propylene coverage, but could be poisoned under high surface coverages. To provide a more comprehensive understanding, we conducted a combination of reactive molecular dynamics (RMD) and classical molecular dynamics (CMD) to simulate the reaction for the $Au_1$ catalyst

as shown in Supplementary Movie 1, and the resulting initial and final snapshots are presented in Supplementary Figs. 25, 26, respectively. As depicted in Fig. 4a, gas molecules undergo strong interactions and accumulate over the catalyst surface, with some molecules even penetrating into the silicalite-1 framework. The statistic concentration profiles shown in Fig. 4b exhibit distinct number density distributions of $H_2$, $O_2$, and $C_3H_6$ along the z-axis. The high-density regions of $O_2$ and $C_3H_6$ are ~2.0 Å away from Au SAs, while that of $H_2$ is around 5.0 Å. This implies that $O_2$ and $C_3H_6$ molecules tend to form a solvation layer over the catalyst surface, which in turn impedes the accessibility of hydrogen molecules to the catalyst surface.

The radial distribution functions (RDFs) of $H_2$, $O_2$, and $C_3H_6$ around Au SAs are further analyzed in Supplementary Fig. 27 to get an in-depth understanding of molecule interactions. The peak at 2.1 Å is associated with the nearest neighbor distance between Au SAs and propylene, whose intensity is much stronger than that of 2.3 Å for Au SAs and oxygen. Moreover, two additional peaks at 2.6 and 3.0 Å can be observed for $C_3H_6$, arising from its adsorption configuration of $\mu_1\eta^1(CH_2)$ in Supplementary Fig. 27. In contrast, hydrogen exhibits the nearest neighbor distance of 2.8 Å with the weakest intensity. These findings suggest the highest probability of finding $C_3H_6$ molecules around Au SAs, which is the lowest for $H_2$. Based on the RDFs results, we select the neighbor distance of 2.25 Å as the statistical criterion for strong chemical adsorption around Au SAs, and studied the dynamic

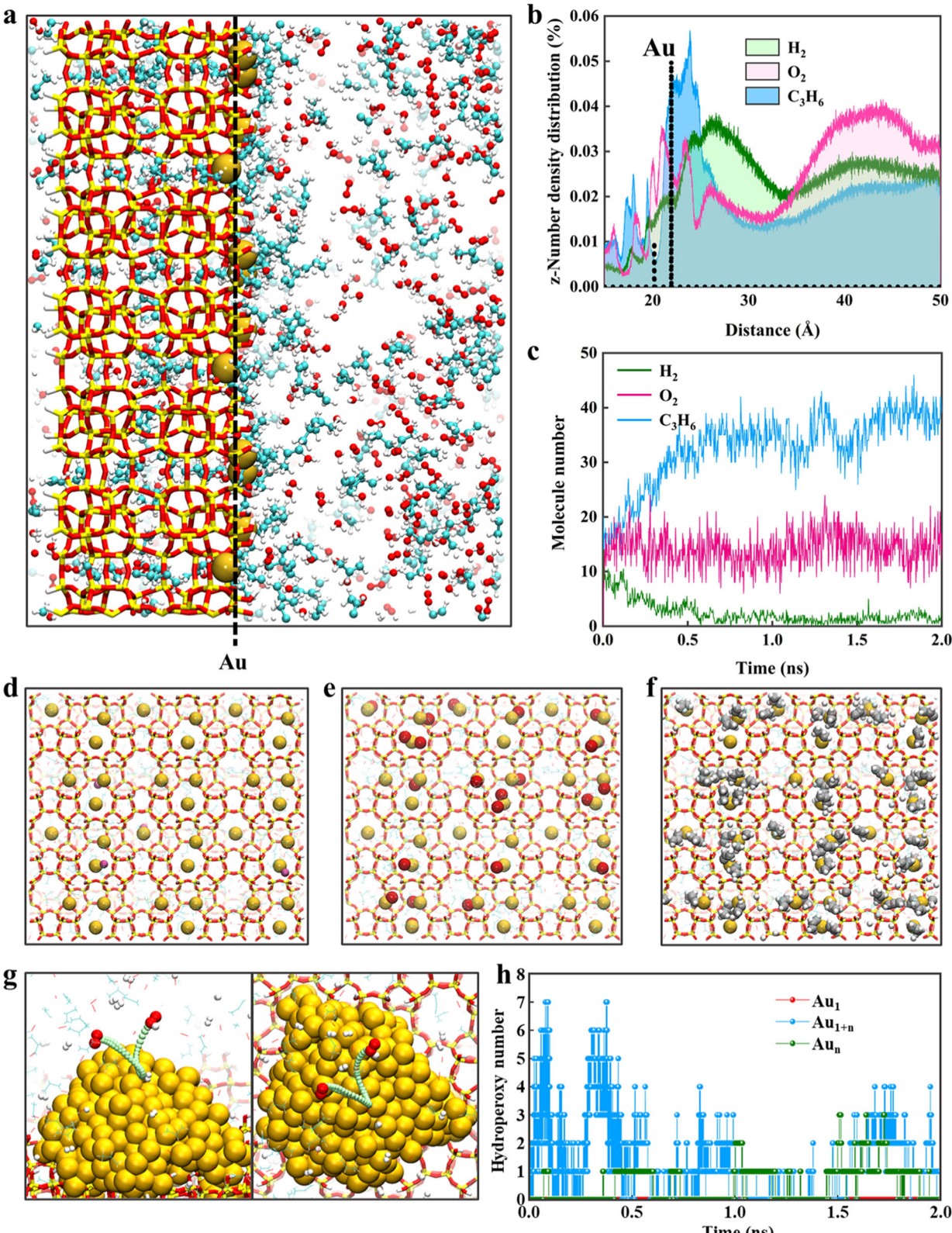

**Fig. 4 | RMD simulation on the poisoning of Au SACs. a, b** Representative snapshot (**a**), and the corresponding number density distributions of $H_2$, $O_2$, and $C_3H_6$ (**b**) over $Au_1$ catalyst along the z-axis. The data is based on the well-equilibrated configuration of a 2.0 ns for the RMD-CMD simulations. **c** The number of adsorbed $H_2$, $O_2$, and $C_3H_6$ over $Au_1$ catalyst as a function of RMD-CMD simulation time. **d–f** Representative snapshots of the adsorption of $H_2$ (**d**), $O_2$ (**e**), and $C_3H_6$ (**f**) on Au SAs at 2.0 ns for the RMD-CMD simulations. **g** The reaction path for the production of hydroperoxy intermediate from $H_2$ and $O_2$ over Au NPs. **h** The number of hydroperoxy intermediate over the $Au_1$, $Au_{1\&n}$, and $Au_n$ catalyst as a function of RMD-CMD simulation time.

adsorption process of different molecules. As shown in Fig. 4c, the numbers of different gas molecules adsorbed on Au SAs were almost the same after a low-temperature relaxation. However, when the temperature rapidly rose to the target temperature of 473 K, the number of adsorbed $H_2$ dropped to almost zero compensated by a significant increase in $C_3H_6$, while that of $O_2$ remained almost unchanged. Consequently, as illustrated in Fig. 4d–f, Au SAs were easily occupied by numerous $C_3H_6$ molecules with negligible $H_2$ adsorption, where only a few $O_2$ molecules can dissociate into adsorbed oxygen atoms. All these results provide a dynamic picture of the poisoning of Au SAs by $C_3H_6$ adsorption.

Furthermore, we have conducted the same simulations for the $Au_{1\&n}$ and $Au_n$ catalyst to get more information about the reaction, and the corresponding initial and final snapshots are presented in Supplementary Figs. 28–31. In comparison to the $Au_1$ catalyst, which was poisoned by $C_3H_6$ with minimal reaction occurring, the presence of Au NPs promotes the formation of hydroperoxyl radical from $H_2$ and $O_2$ as shown in Supplementary Movie 2 and Supplementary Fig. 32. Moreover, the corresponding reaction pathway is displayed in Fig. 4g, where the gaseous $O_2$ molecule approaches the catalyst surface, and reacts with the adsorbed $H_2$ molecule to generate the hydroperoxyl intermediate. The hydroperoxyl intermediate then desorbs from Au NPs to the gas phase for its further conversion, and such a process is consistent with the above DFT calculations. In this regard, a comparison of the number of the generated hydroperoxyl radical was made for the $Au_1$, $Au_{1\&n}$, and $Au_n$ catalyst. As shown in Fig. 4h, the number of hydroperoxyl radical increases with the reaction temperature for the latter two catalysts, particularly for $Au_{1\&n}$, whose number can reach a maximum of 7. Upon reaching the target temperature of 473 K, its number decreases and then fluctuates, indicating a reaction balance between its formation, transfer, and consumption. Conversely, its number remains negligible for the $Au_1$ catalyst throughout the entire process. In this regard, Au NPs can work with Au SAs for the coverage-matching between different reactants to promote the balance between the formation, transfer, and consumption of hydroperoxyl radical.

## Volcano curves and catalytic descriptor

To this point, although DFT calculations predict that Au SAs are highly active for catalyzing propylene epoxidation, kinetic (isotope) studies, and RMD-CMD simulations have shown that they are easily poisoned by the strong adsorption of $C_3H_6$. As a result, $Au_1$ catalyst with a high composition of Au SAs exhibits lower catalytic performances than $Au_n$ catalyst. On the other hand, Au NPs exhibit a weak binding with $H_2$, $O_2$, and $C_3H_6$, allowing the reaction between hydrogen and oxygen to generate the active hydroperoxyl radical. The hydroperoxyl radical can then transfer to adjacent Au SAs and react with the adsorbed $C_3H_6$ to produce PO. From an adsorption point of view, Au NPs can help complete the catalytic cycle by lowering $C_3H_6$ coverage and restoring more active sites. From an activation point of view, Au NPs can synergistically cooperate with Au SAs to provide different types of active sites required for the activation of $H_2/O_2$ and $C_3H_6$, respectively. Therefore, Au NPs can serve as an antidote for Au SAs to overcome the two widely-recognized challenges faced by single-atom catalysts. Combining the merits of both Au SAs and Au NPs, $Au_{1\&n}$ catalyst exhibits the highest catalytic performance for PO production.

It is worth to note that the above $Au_{1\&n}$ catalyst is still dominated by the metallic character of Au NPs rather than the ionized character of Au SAs (Fig. 1g). In this regard, we further tried to maximize Au atom utilization by investigating the intimacy and composition of Au SAs and Au NPs as shown in Fig. 5. At a given composition of $Au_n$ to $Au_1$ ($Au_n$:$Au_1$ = 1:3), it can be seen in Fig. 5a that the catalyst with a dual bed manner, which provides a "millimeter-scale" distance between Au SAs and Au NPs, exhibits a higher catalytic performance than either $Au_1$ or $Au_n$ alone. By further shortening the proximity between $Au_1$ and $Au_n$ to a micrometer-scale distance through mortar mixing, the highest PO

formation rate of 170.1 $mol_{PO} \cdot mol_{Au}^{-1} \cdot h^{-1}$ is achieved, which is >16 times higher than that of the $Au_1$ catalyst. Additionally, the improved epoxidation rate is accompanied by a continuously increased $H_2$ efficiency in Supplementary Table 4, indicating more conversion of hydroperoxyl into PO rather than water. The results suggest that the intimacy between $Au_1$ and $Au_n$ strongly affects the catalytic performance, with long spatial distances making the transfer of hydroperoxyl radical and the subsequent propylene epoxidation difficult. Such mass transfer effect in tandem process was also observed and proposed as the "intimacy criterion", which was first proposed by Weisz on investigating isomerization and hydrocracking[43]. Moreover, the corresponding kinetics study in Supplementary Fig. 33 gives the reaction order of 0.28, 0.31, and 0.35 for $H_2$, $O_2$, and $C_3H_6$, respectively. Obviously, the addition of $Au_n$ into $Au_1$ catalyst can significantly lower the reaction order of $H_2$ compensated with an increase of $C_3H_6$ reaction order. This implies the reduced $C_3H_6$ surface coverage to alleviate its poisoning for more $H_2$ adsorption and reaction.

Tailoring the composition of $Au_1$ and $Au_n$, as a representative of the number of Au SAs and Au NPs, on the micrometer-scale intimacy can further improve the catalytic performance, as depicted by the volcano curve as shown in Fig. 5b and Supplementary Table 5. Surprisingly, the mortar mixing of a few $Au_n$ to $Au_1$ catalyst in a ratio of 1:7 offers a 56-fold improvement in activity (583.6 $mol_{PO} \cdot mol_{Au}^{-1} \cdot h^{-1}$), 3-fold improvement in PO selectivity (75.6%), as well as 22-fold improvement in $H_2$ efficiency (30.5%) compared to $Au_1$ single-atom catalyst. The almost linear correlation between PO formation rate and $H_2$ efficiency as shown in Supplementary Fig. 34 indicates a rate-matching scenario for the formation, transfer, and consumption of hydroperoxyl radical to maximize the catalytic performances. It is worth to note that this PO formation rate even outperforms a majority of Au-Ti bifunctional catalysts in previous studies (Supplementary Table 6). Meanwhile, we have conducted the in-situ SR-FTIR measurement of $C_3H_6$ desorption, in which the band ranging from 2935 to 3015 $cm^{-1}$ was compared in Fig. 5c, where the influences of $C_3H_6$ adsorption on S-1 support could be excluded as shown in Supplementary Fig. 35. Obviously, the substitution of $Au_1$ by $Au_n$ catalyst significantly lowers the amount and strength of $C_3H_6$ adsorption over the mixed catalyst surface. Additionally, we have also investigated the corresponding reaction orders, and the results are illustrated in Supplementary Figs. 36–40 and summarized in Supplementary Table 7. Evidently, the reaction order of $C_3H_6$ increases with the substitution of $Au_1$ by $Au_n$ catalyst, while the trend is completely reversed for $H_2$, further confirming the compromise between $C_3H_6$ and $H_2$ adsorption.

Considering all the catalysts mentioned above, the difference in reaction order between $C_3H_6$ and $H_2$, denoted as $n_{C3H6}$-$n_{H2}$, can serve as an indicator of the competition between $C_3H_6$ and $H_2$ adsorption, and correlate with the catalytic performance in Figs. 6a–c. It can be seen that both the PO formation rate and selectivity as well as $H_2$ efficiency exhibit volcano curves as a function of $n_{C3H6}$-$n_{H2}$, where the reaction is limited by the poisoning adsorption of $C_3H_6$ at the left side and $H_2$ at the right side. As a result, $C_3H_6$ mainly converts into propanal and acrolein over the catalyst surface mainly covered by $C_3H_6$ and $H_2$, respectively. In order to produce PO, the reactant surface coverages can be fine-tuned by the intimacy and composition of Au SAs and Au NPs. Hence, from the perspective of mesokinetics[14], $n_{C3H6}$-$n_{H2}$ is identified as the catalytic descriptor for quantitively correlating the microscopic properties of active sites, including the intimacy and composition of Au SAs and Au NPs, with the macroscopic catalytic performance, which can predict catalytic function and screen catalysts. Accordingly, the mortar-mixing catalyst ($Au_n$:$Au_1$ = 1:7) is suggested to achieve the coverage-matching with moderate $n_{C3H6}$-$n_{H2}$ in Fig. 6a–c, thus contributing to a rate-matching scenario with the highest catalytic performance. The stability of such catalyst was also tested as shown in Fig. 6d, whose reaction rate can be maintained for 150 h without a significant decrease. The observed high stability is

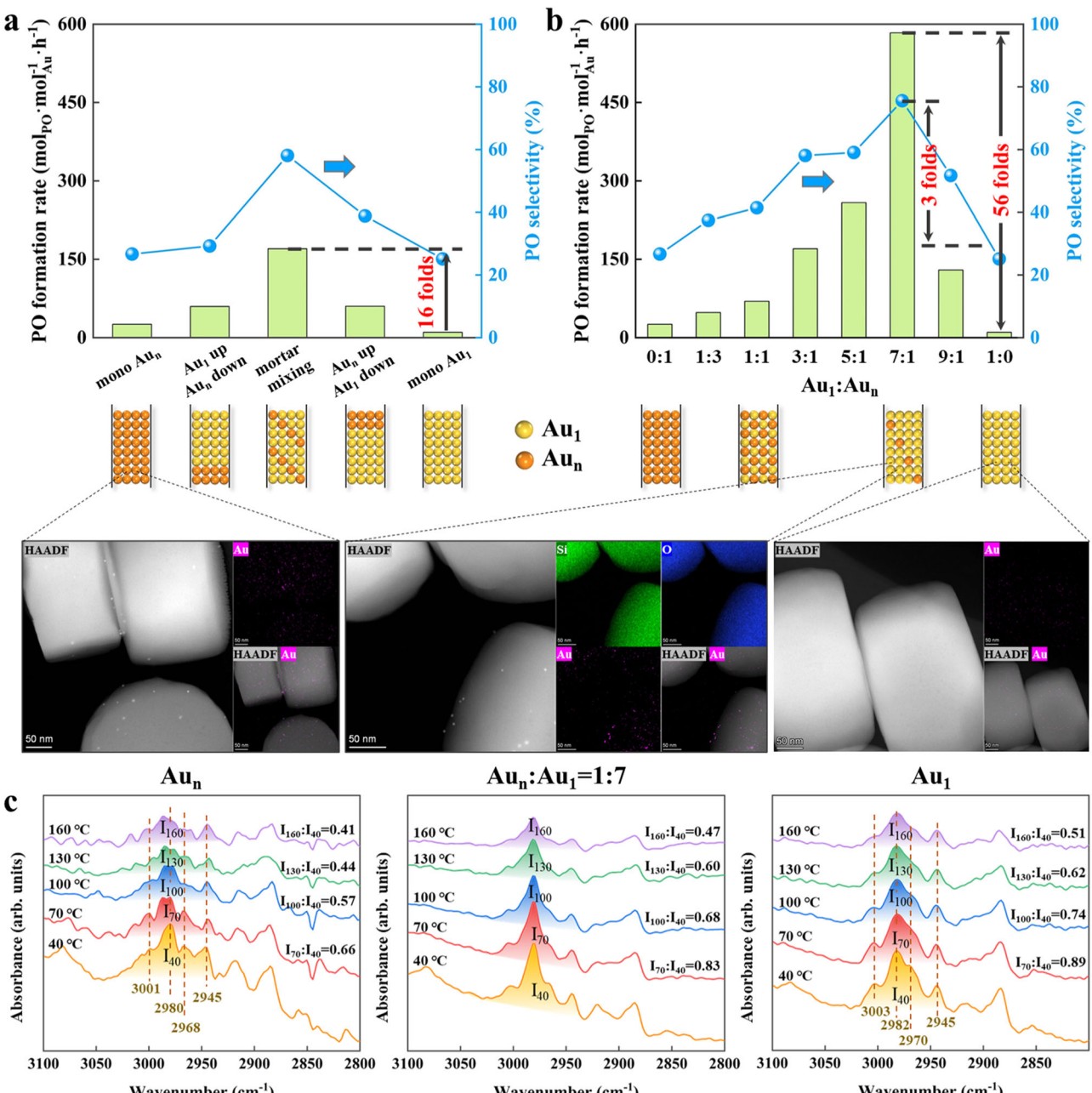

**Fig. 5 | Effects of intimacy and composition of Au SAs and Au NPs. a, b** PO formation rate and selectivity for mixing $Au_1$ and $Au_n$ in different intimacies (**a**) and compositions (**b**). **c** In-situ SR-FTIR spectra of $C_3H_6$ desorption for the $Au_n$, mortar-mixing catalyst ($An_n:Au_1 = 1:7$), and $Au_1$ catalyst at elevated temperatures.

likely due to the robust interactions between Au single atoms and the S-1 support, as depicted in Supplementary Fig. 41, effectively hindering site agglomeration during the reaction.

After the long-term testing, the mortar-mixing catalyst exhibited a gradual decrease in reaction rate, which could be ascribed to the slight increase in Au particle size by particles sintering (Supplementary Figs. 42, 43) and the poisoning of Au active site by carbonaceous deposits (Supplementary Figs. 44, 45). Moreover, the significant shift in Au $4f$ binding energy, as shown in Supplementary Fig. 45, is very likely to have originated from the decomposition of tetra-propylammonium hydroxide (TPAOH), which serves as the structure-directing agent for S-1. Furthermore, for the mortar-mixing catalyst ($Au_n:Au_1 = 1:7$), the number ratio of Au SAs to Au NPs can be calculated based on our previous study as 308:1 as seen in the Supplementary Information[44]. Hence, as shown in Fig. 6e, Au NPs can serve as antidote of Au SACs with only trace addition (0.3% ratio of Au SACs) to

simultaneously improve the PO formation rate from 10.4 to 583.6 $mol_{PO} \cdot mol_{Au}^{-1} \cdot h^{-1}$, PO selectivity from 25.1% to 75.6%, and $H_2$ efficiency from 1.4% – 30.5%, offering a 56-fold, 3-fold, and 22-fold increase, respectively, whose performances can be maintained for 150 h. It is worth noting that these catalysts still encounter certain issues, such as a prolonged induction period and inadequate conversion at low Au loadings. These challenges can potentially be mitigated by selecting calcined S-1 as the catalyst support for achieving precise atomic-level synthesis of Au/S-1 subnanocatalysts with a higher Au loading.

## Discussion

In this study, we conduct a proof-of-concept study on the poisoning of Au SACs, and the antidote of Au NPs with only trace addition (0.3% ratio of Au SACs) to reinforce and sustain propylene epoxidation. Although theoretical calculations predict the remarkable epoxidation activity of Au SACs, experimental investigations suggest the

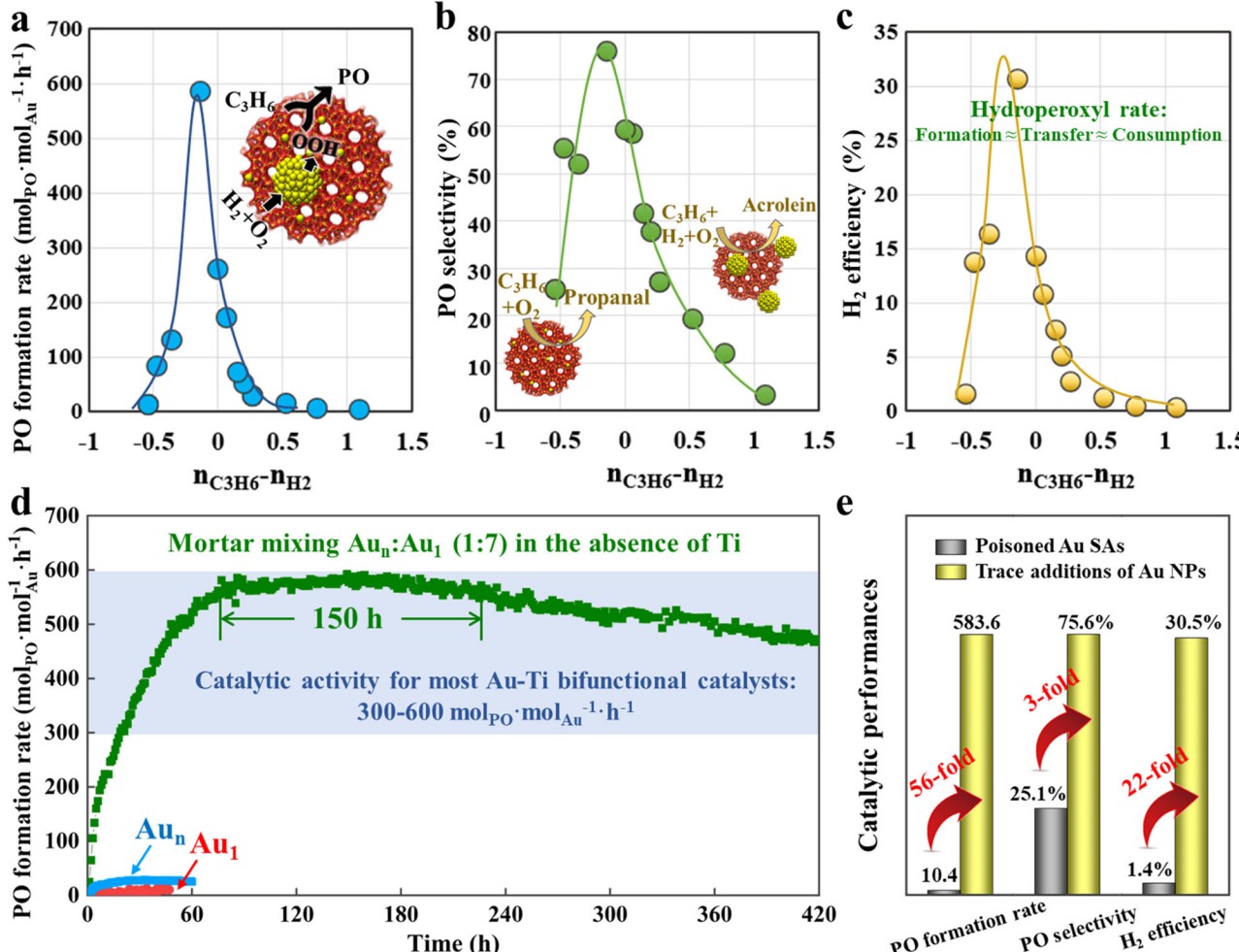

**Fig. 6 | Volcano curves and catalytic performances comparison. a–c,** PO formation rate (**a**) and selectivity (**b**), as well as $H_2$ efficiency (**c**) as a function of $n_{C3H6}$-$n_{H2}$. **d** Stability test of mortar mixing $Au_1$ and $Au_n$ in the $Au_n$/$Au_1$ mass ratio of 1:7 for propylene epoxidation. **e** Catalytic performances comparison between poisoned Au SAs and those with trace addition of Au NPs (0.3% ratio of Au SACs).

completely opposite kinetics behaviors, including activation energy, reaction orders, and kinetic isotope effects. Further combining multiscale simulations (DFT and RMD-CMD) formulates a dynamic insight into this contradictory phenomenon, where Au SAs exhibit high catalytic activity at a low $C_3H_6$ surface coverage, but become poisoned by a solvation layer under a high coverage. In contrast, Au NPs with a weaker binding to $C_3H_6$ can effectively catalyze $H_2$ and $O_2$ to generate the key intermediates of hydroperoxyl radical that can transfer and react with $C_3H_6$ on Au SAs. Hence, Au NPs can not only help complete the catalytic cycle by reducing $C_3H_6$ coverage and restoring Au SAs, but also synergistically cooperate with Au SAs by providing different types of active sites required for the activation of $H_2$/$O_2$ and $C_3H_6$. Thus, Au NPs can serve as the antidote for Au SAs to alleviate their poisoning for sustainable PO production, providing the highest catalytic performance of 80.2 $mol_{PO}\cdot mol_{Au}^{-1}\cdot h^{-1}$ of $Au_{1\&n}$ catalyst than the $Au_1$ (10.4 $mol_{PO}\cdot mol_{Au}^{-1}\cdot h^{-1}$) and $Au_n$ (25.9 $mol_{PO}\cdot mol_{Au}^{-1}\cdot h^{-1}$) catalyst.

A further advancement is made by identifying the difference in reaction order between $C_3H_6$ to $H_2$ ($n_{C3H6}$-$n_{H2}$) as the catalytic descriptor to establish the volcano curves. It is demonstrated that the reaction is limited by the poisoning adsorption of $C_3H_6$ at the left side of the volcano curves to produce propanal, while that of $H_2$ at the right side to yield acrolein. Interestingly, $n_{C3H6}$-$n_{H2}$ can be fine-tuned by the intimacy and composition of Au SAs and Au NPs to afford the surface coverage-matching between reactants, thus achieving a rate-matching scenario for the formation, transfer, and consumption of hydroperoxyl

radical. As a result, only trace addition of Au NPs (0.3% ratio of Au SACs) enables a simultaneous increase in PO formation rate from 10.4 to 583.6 $mol_{PO}\cdot mol_{Au}^{-1}\cdot h^{-1}$, PO selectivity from 25.1% to 75.6%, and $H_2$ efficiency from 1.4% to 30.5%, which can be maintained for 150 h. This proof-of-concept study, by introducing trace amounts of antidote to counteract the poisoning of SACs and even reinforce their catalytic performances, demonstrates great potential in accelerating the industrial applications of SACs for a sustainable chemicals production.

## Methods
### Catalyst preparation
The Ti-free silicalite-1 (S-1) was employed as catalyst support, which was prepared by the hydrothermal method. Typically, 2.00 g polyoxyethylene 20-sorbitan monolaurate (Tween 20, Sinopharm Chemical Reagent Co., Ltd.) was firstly dissolved in 28 mL ultrapure water, followed by the dropwise addition of 23.00 g TPAOH (25 wt% in water, Shanghai Mindray Chemical Technology Co., Ltd.) under vigorous stirring for 1 h and then 40.53 g tetraethylorthosilicate (TEOS, Shanghai McLean biochemical technology Co., Ltd.) for another 1 h. The as-obtained solution was transferred to a Teflon autoclave, which was heated at 443 K for 48 h under autogenous pressure for crystallization. The crystalline solid was separated from the slurry by centrifugation, subsequently washed and dried at 373 K for 12 h, and the as-obtained sample was directly used as the catalyst support without calcination.

The Au/S-1 catalysts were prepared by the deposition-precipitation-urea method. Typically, 1.00 g S-1 support was mixed with 40.00 mL deionized water under vigorous stirring. Then, a certain amount of 4.85 mM chloroauric acid solution (HAuCl$_4$·4H$_2$O, Sinopharm Chemical Reagent Co., Ltd.) and urea (Shanghai Aladdin Biochemical Technology Co., Ltd.) was added into the above solution, where the molar ratio of chloroauric acid to urea was kept as 1:100. Moreover, the amount of chloroauric acid solution varies from 0.4 mL to 6.0 mL to obtain Au/S-1 catalysts with different sizes. After heating to 363 K under vigorous stirring for another 6 h, the as-obtained catalyst precursor was separated from the slurry by centrifugation, subsequently washed and dried at room temperature overnight under vacuum.

## Catalyst characterization

Inductively coupled plasma-atomic emission spectrometry (ICP-AES) analysis was performed on a Varian 710-ES instrument to determined Au loading. X-ray diffraction (XRD) patterns were recorded on a Rigaku D/max2550VB/PC X-ray diffractometer with Cu Kα radiation. High-angle annular dark-field scanning transmission electron microscopy (HAADF-STEM) and element energy-dispersive spectroscopy (EDS) mapping measurements were performed on the ThermoFisher Talos F200X microscope at an operating voltage of 200 kV. Aberration-corrected scanning transmission electron microscopy (AC-STEM) was operated on a ThermoFisher Themis Z transmission electron microscope equipped with two aberration correctors at an operating voltage of 300 kV. To minimize the electron beam damage on zeolite frameworks, a low-damage high angle annular dark field (HAADF)-STEM imaging technique was employed[45]. X-ray photoelectron spectra (XPS) measurements were obtained on an ESCALAB 250 Xi, ThermoFisher with Mg/Al Kα radiation, where all the binding energies were calibrated by the C 1s peak at 284.6 eV. In-situ synchrotron-based FTIR (SR-FTIR) of C$_3$H$_6$ adsorption and desorption experiments were performed on a Nicolet 6700 FTIR Spectrometer with Nicolet Continuum IR Microscope in transmission mode in the Shanghai Synchrotron Radiation Facility (SSRF), Shanghai, China. Firstly, the sample was pretreated under the gas flow (O$_2$:H$_2$:C$_3$H$_6$:N$_2$ = 1:1:1:7) of 30 mL min$^{-1}$ at 200 °C for 1 h. Then, the sample was purged by N$_2$ (30 mL min$^{-1}$) and cooled down to 40 °C. After the spectra were stable, the background was collected. Afterwards, the gas was switched to the mixture (H$_2$:C$_3$H$_6$:N$_2$ = 1:1:1:7) with the flow rate of 30 mL min$^{-1}$ for 20 min to ensure the saturated adsorption, and further switched back to N$_2$ (30 mL min$^{-1}$). Finally, the data was collected after the spectra were stable, during which the temperature was increased gradually to 160 °C. The spectra collected at different temperature were subtracted from background spectra at the corresponding temperature. In-situ DRIFTS spectra were collected in an in-situ diffuse reflection cell (Harrick Praying Mantis) placed in PerkinElmer Spectrum 100 FTIR spectrometer, in which the spectra were subtracted from the background spectra.

## Catalyst testing

The propylene epoxidation reaction using H$_2$ and O$_2$ was conducted in a fixed bed quartz tubular reactor with an inside diameter of 8 mm under atmospheric pressure. A catalyst weighing 0.15 g and sieved to 200 mesh size was loaded into the reactor and reduced by a H$_2$/N$_2$ mixture (2/3 by volume) with a flow rate of 42 mL·min$^{-1}$ at 583 K for 1.5 h. After cooling the catalyst to 473 K under N$_2$, the reactant mixture containing O$_2$, H$_2$, C$_3$H$_6$, and N$_2$ was introduced with flow rates of 3.5, 3.5, 3.5, and 24.5 mL•min$^{-1}$, respectively, and a space velocity of 14000 mL·h$^{-1}$·g$_{cat}$$^{-1}$. The reactant and product components were analyzed using an on-line online mass spectrum (MS) and gas chromatograph (GC) equipped with a TCD detector and FID detector. Oxygenates, such as propylene, PO, acetone, acrolein, and propanal, were separated using a RT-QS-Bond capillary column (0.35 mm × 30 m) and analyzed with an FID detector. The H$_2$, O$_2$, and CO$_2$ were separated using a TDX-01 column (3 mm × 3 m) and analyzed with a TCD detector.

## DFT calculations

All DFT calculations were performed using the Vienna Ab initio Simulation Package (VASP) code[46–48]. The electronic exchange and correlation energy were calculated employing the generalized gradient approximation proposed by Perdew-Burke-Ernzerhof (GGA-PBE)[49]. The projected augmented wave method (PAW) was used to describe the interaction between the ionic cores and valence electrons, with a cut-off energy of 400 eV[50]. The DFT-D3 correction scheme was employed to assess the weak van der Waals interactions between the adsorbed molecules and the catalyst surface[51]. A Monkhorst-Pack grid was used for k-point sampling in the Brillouin zone[52]. Specifically, a 3 × 3 × 1 Monkhorst-Pack k-point mesh within the surface Brillouin zones was used for Au(111) model, while that of 1 × 1 × 1 Monkhorst-Pack k-point mesh for Au single atom and Au$_{13}$ cluster models. A force-based conjugate gradient method was used for geometry optimizations. The transition states of the elementary steps were located by means of the dimer method, and vibrational analyses were further carried out to ensure the local minima and transition states[53]. A threshold of 10$^{-5}$ eV was employed for the convergence of the electronic structure. Convergence of saddle points and minima were believed to reach when the maximum force in each degree of freedom was <0.03 eV · Å$^{-1}$.

## Hybrid classical MD technique and ReaxFF method

A combined reactive force field (ReaxFF) molecular dynamics and classic molecular dynamic simulations were performed using the LAMMPS simulation package[54–56]. Regarding the inert catalyst support of S-1 and the main active species of Au, we used a classical force field to describe the interaction forces between the reactants and Au as well as S-1. To capture the adsorption, diffusion, and reaction behavior, we utilized an Au/C/H/O reaction force field that was optimized based on data obtained from density functional theory (DFT) calculations, which provides a more accurate representation of the Au system and its interactions with carbon, hydrogen, and oxygen. To describe the interactions within the S-1 framework, we employed the Tersoff potential function. This potential function is specifically designed to investigate the structural properties of Si-O systems, making it suitable for modeling the interactions within S-1[57]. Moreover, we have incorporated the two-body Lennard-Jones (LJ) potentials with a cutoff distance of 12 Å to describe the interaction between the S-1 framework and Au catalyst as well as gas molecules. The form of the potential can be written as:

$$\phi_{ij}(r) = \frac{q_i q_j}{r_{ij}} + 4\varepsilon \left[ \left( \frac{\sigma}{r_{ij}} \right)^{12} - \left( \frac{\sigma}{r_{ij}} \right)^6 \right] \tag{1}$$

where the corresponding potential parameters can be adopted from the literatures, and summarized in Supplementary Table 8[58,59]. The force field (Au/C/H/O ReaxFF) adapted from the work of Monti et.al.[60]. was applied to properly describe the bond dissociation and formation processes, which can precisely describe the anchoring mechanism of cysteine and gold in an aqueous solution.

## Data availability

The authors declare that all the important data to support the findings in this paper are available within the main text or in the Supplementary information. Extra data are available from the corresponding author upon request.

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

## Acknowledgements

W.C., X.Z. and X.D. acknowledge funding from the National Key R&D Program of China (2021YFA1501403), the Natural Science Foundation of China (22038003, 22178100, 22178101, U22B20141 and 22008066), the Shanghai Pilot Program for Basic Research (22TQ1400100-15), the Innovation Program of Shanghai Municipal Education Commission, the Program of Shanghai Academic/Technology Research Leader (21XD1421000), the Shanghai Science and Technology Innovation Action Plan (22JC1403800). The authors thank the staff members from the BL06B beamline of Shanghai Synchrotron Radiation Facility (SSRF) for assistance during data collection.

## Author contributions

W.C., X.Z. and X.D. conceived this work. W.C., Q.W. and K.S. performed the experiments, collected the data, and wrote the paper. X.Z. and X.D. designed the research, supervised experiments, and edited the paper. C.Liu and C.Lian conducted the density-functional theory calculation and molecular dynamic simulations. T.J. and L.L. helped with in-situ SR-FTIR analyses. Z.Z., G.Q. and J.Z. assisted in the HAADF-STEM and XPS characterization. W.Y. helped with data analyses and discussions. All the authors contributed to the paper revisions.

## Competing interests

The authors declare no competing interests.
