## [Peer Review File · Nature Communications]

Nanoparticles as an Antidote for Poisoned Gold Single-Atom Catalysts in Sustainable Propylene EpoxidationREVIEWER COMMENTS

Reviewer #1 (Remarks to the Author):

In the submission the authors reported an interesting observation of synergistic effects of Au SACs and Au NPs supported on S-1 for propylene epoxidation with H₂ and O₂. Although such an observation was novel and interesting, I did not consider that they were qualified to be published in Nat. Commun. due to the following issues:

1. The comparisons of catalytic performances to previous results in molPO/molAu were not fair. The authors should present C₃H₆ conversions in the manuscript. As shown in Table S6, the catalytic performance of the reported catalyst was much poorer than most Au/TS-1 catalysts with careful preparations. Thus, the conclusion "Such performances can be maintained for 150 hours and even outperform most widely-used Au-Ti bifunctional catalysts." could not be held.
2. As shown in Figure 6d, the reported catalyst seemed to have a long induction period during the catalytic reaction, which indicated that the Au SACs should not be good for catalyzing the reaction.
3. Figure 1g, why was the Au 4f_{7/2} binding energy of the catalyst with the highest Au loadings only 83.2 eV, much smaller than that of bulk metallic Au? The authors were suggested to compare the Au XANES spectrum with the standard one.
4. Figure 1i: did the authors exclude the possibility of C₃H₆ adsorption on the support during the in situ FTIR measurements? Were any differences in the C=C vibrational features observed? The S/N ratios of the FTIR spectra were poor.
5. How were the C₃H₆ conversions measured, directly by the C₃H₆ concentration changes or indirectly by calculations from the products? If the latter, errors could be quite large on the kinetic data.

Reviewer #2 (Remarks to the Author):

Summary:

Wang et al. performed a comprehensive proof-of-concept study on the catalytic performance of silicalite-supported Au single atoms (SAs), nanoparticles (NPs), and SA/NP mixes for propylene epoxidation. It was found that at intermediate loadings of Au (0.014 wt%) resulted in dramatically improved PO formation rates. (S)TEM imaging suggests that these highly active samples contain a mixture of Au NPs and SAs. The authors hypothesize that Au SAs have favorable reaction kinetics at low C₃H₆ coverages, but become poisoned at high coverages. However, the introduction of trace Au NPs provided the "antidote" to this poisoning by providing sites for hydroperoxyl radicals to form. This hypothesis is supported by a combination of DFT and MD simulations, as well as catalyst mixing tests. The authors also provide mechanistic insights and reaction profiles supported by computation. Finally, the authors provide a descriptor to determine optimal balance between NP and SA sites based on the difference between C₃H₆ and H₂ reaction order.

General Comments:

I recommend that the scope, impact, and quality of this paper are appropriate for Nature

Communications, though there are some concerns which should be addressed before publishing. While the title and abstract are written somewhat confusingly, the rest of the manuscript is easy to follow. The authors have provided a strong combination of experimental and computational studies which support their hypothesis, and this work presents a new avenue for studying synergistic effects between metal NPs and SAs.

Specific Major Questions/Concerns:

1. The use of the word “antidote” in the title and abstract is somewhat confusing, particularly because poisoning isn’t mentioned in the title. It’s longer, but something along these lines might be easier to understand: “Gold nanoparticles provide the antidote for single atom catalyst poisoning during sustainable propylene epoxidation in the absence of titanium” or “Synergistic effects between Au single atoms and nanoparticles lead to enhanced sustainable propylene epoxidation in the absence of titanium”
2. The analytical methodology used for particle size quantification should be made more clear. How many samples/particles/images were used to determine average particle size? Can you also clarify how uniform the size distribution of Au NPs was, or provide a particle size histogram? Providing these details as well as additional (S)TEM images would strengthen claims made about the structure of the catalysts.

Specific Minor Questions/Comments:

1. It would be interesting if the authors provided insights on how an optimal ratio of Au SAs and NPs could be achieved at higher metal weight loadings
2. While the on-stream stability looks generally good, can the authors comment on the loss in activity over 100s of hours in Fig 6d? Do they believe it’s due to poisoning, or perhaps catalyst sintering?
3. Are the normalized PA formation rates, such as those in Figure 5a, calculated based on total Au content or Au surface sites for samples with Au NPs (i.e. is decreased dispersion accounted for)?

Reviewer #3 (Remarks to the Author):

Overall this is a good and thoroughly thought paper. It provides some very good ideas in the behaviour of Single Atom Catalysts in regard to a technically (in terms of reaction mechanism) challenging and commercially important reaction. I believe the paper provides various new insights in the behaviour of Au atoms and clusters. I think the paper is suitable for publication in Nature Communication. However the authors should take into account the following comments (especially comments 6).

1. I think the introduction (first paragraph) where the overall behaviour and characteristics of Single Atom Catalysts is described requires more of a recent review type of reference like <https://doi.org/10.1016/j.pecs.2023.101074> or <https://doi.org/10.1016/j.pecs.2023.101074>
2. Figure S1 is a nice and useful figure showing the energy profiles of the two pathways under consideration. However, the small images the authors have inserted are hard to see (at least with the

image resolution provided in the reviewer document I have received). Please make sure that the resolution of the images is improved. (The same applies for Figure S2),

3. In connection with the previous comment, the two mechanisms are very important for the manuscript. The authors may wish to consider to include them in the form of simple chemical equations in the main manuscript. (I understand that this may not be possible, this is mostly a suggestion).

4. Figure 1 caption, I think it would help a lot if the authors could add the gold loading in parenthesis next to Au1, Au1&n and Au_n.

5. In regard to the XPS, I agree that the Au4f shift is most likely related to final state effects, however the authors should add a relevant reference (there are plenty in the literature, so I am not suggesting a specific one).

6. Looking overall at the manuscript, I am not sure how the authors prove that they have obtained a Single Atom Catalyst (or at least this is not well explained). From HAADF-STEM it is not clear that we are dealing with single atoms. Possibly NEXAFs could tell more (from the point of view that from the EXAFS equation they can obtain the coordination number). How are they sure that they are not dealing with Au dimers or trimers. Very likely the low loading material (Au1) contains all the above (tiny clusters of Au and possibly single atoms too). I think this has to become very clear in the manuscript.

7. I believe that the manuscript would benefit with some more discussion on the post reaction characterization of the materials used.

Reviewer #4 (Remarks to the Author):

The manuscript by Wang, Sang, Liu et al. presents interesting and novel results based on a sophisticated methodology in the search for a new catalyst for propylene epoxidation that avoids the problems of the single Au atom catalyst supported on silicalite-1 and Ti-silicalite-1. The catalyst suggested by the authors is based on the mix of Au NPs and Au SAC that avoids the poisoning of Au SAC by generating hydroperoxyl on the Au NPs. The experimental results suggest the increase in the catalytic activity of the mixed catalyst while the characterization of the material confirms the existence of SAC and NPs by means of a method that mixes different weight loads. The theoretical counterpart gives solid evidence that contributes to the understanding of the mechanisms that allow the mixed catalyst to improve the catalytic activity of the system. The results are very significant in the field, and I recommend widely its publication in Nature Communications. However, there are some questions that could help to clarify the synergetic effect of the Au NPs and Au SA in the mixed catalyst. The issues that could help to improve the quality of the manuscript are the following:

1) The diffusion barrier of Au atoms on silicalite-1 could be calculated by means of DFT calculations to know the capability of Au atoms to sinter and create larger NPs and reduce the catalytic activity of the mixed catalyst.

2) The MD without the molecules could also show the possibility of the sintering process.

3) The contribution of the vdW forces in the molecule-Au interaction is very important. Could you separate that contribution to know how big it is? Could you do a single-point or relaxation calculation with a more accurate method to see that the results are consistent and do not depend on the vdW

forces approximation?

4) The temperature in the MD simulation is not explained in detail. Is the temperature constant or is there a temperature ramp? What is the size of the step? Why that REAXFF potential is employed? Could you include the comparison of adsorption energies between REAXFF potential and DFT?

5) The author talks about the ionized character of Au SA. The theoretical calculation of the charge distribution (NBO, Bader, or other method) or the shift 4f states could corroborate this result.

Many thanks for the valuable comments and suggestions from the four reviewers. We have revised our paper by fully taking into account all the comments and suggestions.

Reviewer #1 (Remarks to the Author):

In the submission the authors reported an interesting observation of synergistic effects of Au SACs and Au NPs supported on S-1 for propylene epoxidation with H₂ and O₂. Although such an observation was novel and interesting, I did not consider that they were qualified to be published in Nat. Commun. due to the following issues:

1. The comparisons of catalytic performances to previous results in mol_{PO}/mol_{Au} were not fair. The authors should present C₃H₆ conversions in the manuscript. As shown in Table S6, the catalytic performance of the reported catalyst was much poorer than most Au/TS-1 catalysts with careful preparations. Thus, the conclusion "Such performances can be maintained for 150 hours and even outperform most widely-used Au-Ti bifunctional catalysts." could not be held.

Response:

We fully agree with the referee that in addition to PO formation rate, selectivity, and H₂ efficiency, C₃H₆ conversion is another important criterion for catalyst evaluation. Accordingly, we have made a comparison of PO formation rate, selectivity, H₂ efficiency, and C₃H₆ conversion in Table S6. It can be seen that the Au₁&Au_n catalyst exhibits comparable and even better PO formation rate, selectivity, and H₂ efficiency as Au-Ti bifunctional catalysts. However, it is worth noting that these catalysts still encounter certain issues, such as inadequate conversion due to their low Au loadings. Such problem can potentially be mitigated by selecting uncalcined S-1 as the catalyst support for achieving precise atomic-level synthesis of Au/S-1 subnanocatalysts with a higher Au loading. Accordingly, we have removed the conclusion of "Such performances can be maintained for 150 hours and even outperform most widely-used Au-Ti bifunctional catalysts.", and added the relevant discussion as follows:

"Hence, as shown in Fig. 6e, Au NPs can serve as antidote of Au SACs with only trace addition (0.3% ratio of Au SACs) to simultaneously improve the PO formation rate from 10.4 to 583.6 mol_{PO}·mol_{Au}⁻¹·h⁻¹, PO selectivity from 25.1% to 75.6%, and H₂ efficiency from 1.4% to 30.5%, offering a 56-fold, 3-fold, and 22-fold increase, respectively, whose performances can be maintained for 150 hours. It is worth noting that these catalysts still encounter certain issues, such as a prolonged induction period and inadequate conversion at low Au loadings. These

challenges can potentially be mitigated by selecting calcined S-1 as the catalyst support for achieving precise atomic-level synthesis of Au/S-1 subnanocatalysts with a higher Au loading.”

“Consequently, only trace addition of Au NPs antidote (0.3% ratio of SACs) stimulates significant improvements in propylene oxide formation rate, selectivity, and H₂ efficiency compared to SACs alone, offering a 56-fold, 3-fold, and 22-fold increase, respectively, whose performances can be maintained for 150 hours.”

“As a result, only trace addition of Au NPs (0.3% ratio of Au SACs) can stimulate a simultaneous increase in PO formation rate from 10.4 to 583.6 mol_{PO}·mol_{Au}⁻¹·h⁻¹, PO selectivity from 25.1% to 75.6%, and H₂ efficiency from 1.4% to 30.5%, whose catalytic performances can be maintained for 150 hours.”

Table S6. The comparison of PO formation rate between the monometallic Au catalyst in the current study and bifunctional Au-Ti catalysts in previous studies.

Sample	PO formation rate (mol _{PO} ·h ⁻¹ ·mol _{Au} ⁻¹)	PO selectivity	H ₂ efficiency	C ₃ H ₆ conversion	Reference
Au/U-TS-1	2034.9	92.4	55	2.0	J. Catal. 2014, 313, 104–112
Au/TS-1	1800.0				J. Catal. 2013, 308, 98–113
Au/TS-1	1695.7				J. Catal. 2012, 287, 178–189
Au/TS-1-B	994.8	95	30.9		Appl. Catal. B 2022, 319, 121837
Au/TS-1-B-DPA	847.8	89.6	40	3.2	J. Catal. 2022, 416, 410–422
Au/TS-1	695.2	83	26		ACS Catal. 2018, 8, 10649–10657
Au₁&Au_n	583.6	75.6	30.5	0.29	This work
Au/TS-1	561.0	76	30	10	J. Catal. 2005, 232, 38–42
Au/TS-1	552.8		<30		J. Catal. 2015, 325, 128–135
Au/TS-1-B	540.0	89.6	33.6		J. Catal. 2014, 317, 99–104
Au/HTS-1(NIMG)	509.7		29.1		ACS Sustainable Chem. Eng. 2022, 10, 9515–9524
Au/TS-1-B	508.7	92	35		Ind. Eng. Chem. Res. 2019, 58, 17300–17307
Au/S-1/TS-1@dendritic-SiO ₂	486.3	93.9	26.1		Green Energy Environ. 2020, 5, 473–483

Au/STS-1	478.2	91.2			Catal. Today . 2020, 347, 102–109
Au/TS-1-cS	474.8	90.7	43.6		Engineering 2023, DOI: 10.1016/j.eng.2023.01.008
Au/TS-2-B	444.6	90	35		AIChE J 2020, 66, e16815
Au/HTS-1	423.9	85	~20		ACS Appl. Mater. Interfaces 2021, 13, 26134–26142
Au/TS-1-B	412.2	83			Chem. Eng. J. 2015, 278, 234–239
Au/Ti-SiO₂	410.4	91.9	14	4.8	J. Catal. 2016, 344, 434–444
Au/TS-1	387.1				J. Catal. 2018, 365, 105–114
Au/Ti-SiO₂	376.8	90	14.5		J. Catal. 2016, 338, 284–294
Au/MTS-1	370.4	95.2			ACS Catal. 2017, 7, 2668–2675
Au/TS-1-B	353.3	83	22		Appl. Catal. B 2014, 150–151, 396–401
Au/TS-1-B	352.7	85	22	4.3	ACS Catal. 2018, 8, 7799–7808
Au/TS-1(SG)	215.4	84	24	8.3	J. Catal. 2011, 278, 8–15
Au/S-1/TS-1	195.9				J. Catal. 2012, 296, 31–42
Au/Ge-TS-1	159.9	91		4	J. Catal. 2009, 267, 202–206
Au-Ti@MFI	139.0	83.9	23	14.8	J. Mater. Chem. A 2020, 8, 4428–4436
Au/Mg-TS-1	119.4	85.8		5.7	Catal. Today 2009, 147, 186–195
Au/TS-1	81.5	72.1	27.6	10.1	J. Catal. 2011, 283, 192–201
Au/TiO₂@SBA-15	70.5	62	7.5	2.2	J. Catal. 2011, 282, 94–102
Au/TS-1	27.4	77.2		1.1	J Solid State Chem. 2018, 261, 92–102
Au-PVP/TS-1	21.6				ACS Catal. 2022, 12, 16, 10147–10160
Au/TS-1(PT)	7.6	88	23	1.3	Angew. Chem. Int. Ed. 2021, 133, 18333–18341

2. As shown in Figure 6d, the reported catalyst seemed to have a long induction period during the catalytic reaction, which indicated that the Au SACS should not be good for catalyzing the reaction.

Response:

Thank you for pointing out this important issue. Actually, in our previous study (*AIChE J.*, 2020, 66, e16815), we found that the long induction period can be related to the decomposition of the template absorbed on the external surfaces of the uncalcined titanium silicates. In our current work, we have opted for uncalcined S-1 as the support material in the presence of a template.

This choice serves a dual purpose, as it not only helps in preventing catalyst deactivation due to pore blockage from carbonaceous deposits but also enables the collection of particle size statistical data. Taking inspiration from the referee's suggestion, we can consider using calcined S-1 without a template as the catalyst support to shorten the induction period in our future research. As such, we have added the relevant discussion in the revised version as follows:

“It is worth noting that these catalysts still encounter certain issues, such as a prolonged induction period and inadequate conversion at low Au loadings. These challenges can potentially be mitigated by selecting calcined S-1 as the catalyst support for achieving precise atomic-level synthesis of Au/S-1 subnanocatalysts with a higher Au loading.”

3. Figure 1g, why was the Au 4f_{7/2} binding energy of the catalyst with the highest Au loadings only 83.2 eV, much smaller than that of bulk metallic Au? The authors were suggested to compare the Au XANES spectrum with the standard one.

Response:

As recommended by the referee, it is well-established that bulk gold (Au) atoms possess a binding energy (BE) of 84.0 eV for the Au 4f_{7/2} level, which is significantly higher than that of current catalysts. Our XPS results indicate a notable negative shift in the Au 4f binding energy for nanoparticles supported on S-1, with this shift becoming more pronounced as the catalyst loading increases. This negative shift can be attributed to the transfer of electrons from the S-1 support to the gold particles, resulting in the formation of anionic gold species. It is worth noting that these anionic gold species are considered as the active components for the process of propylene epoxidation (*J. Mater. Chem. A*, **2020**, 8, 4428–4436; *Ind. Eng. Chem. Res.*, **2019**, 58, 17300–17307). They promote the weak adsorption of oxygen, which is crucial for the generation of hydroperoxyl species. Additionally, in an effort to characterize the electronic properties of our catalysts, we attempted X-ray Absorption Spectroscopy (XAS). Unfortunately, the ultralow loading of gold in these catalysts hindered our ability to obtain more detailed information through XAS analysis. In light of these challenges, we have revised our discussion as follows:

“The XPS spectra in Fig. 1g exhibit a continuous increase of Au 4f binding energy by decreasing Au size, most likely attributed to the final state effect resulting from the size-

dependent electrostatic interaction between ionized cluster and escaping photoelectron.³⁶ Moreover, the negative shift of Au 4f binding energy of these catalyst compared with bulk Au indicates electron transfer from S-1 to Au nanoparticles for these catalysts, and the resultant electron-rich Au species has been suggested to weaken oxygen adsorption to promote this reaction.³⁷ Correspondingly, there is a significant decrease in signal intensity, particularly for Au₁, whose signal was too weak to be detectable. Notably, although there are some isolated Au SAs in the Au_{1&n} catalyst, most of Au atoms are ensembled, and the metallic character still dominates, similar to the Au_n catalyst. Unfortunately, XAS failed to gain more insights into their electronic structures and coordination environments due to the low Au loadings.”

37. Wang, L. et al. Titanium silicalite-1 zeolite encapsulating Au particles as a catalyst for vapor phase propylene epoxidation with H₂/O₂: a matter of Au–Ti synergic interaction. *J. Mater. Chem. A* **8**, 4428–4436 (2020).

4. Figure 1i: did the authors exclude the possibility of C₃H₆ adsorption on the support during the in situ FTIR measurements? Were any differences in the C=C vibrational features observed? The S/N ratios of the FTIR spectra were poor.

Response:

We are very sorry for neglecting this important issue. Accordingly, we have conducted the in-situ SR-FTIR measurement of C₃H₆ adsorption over S-1 support in the absence of Au, and the results are shown in Fig. S35. It can be seen that the adsorption of C₃H₆ adsorption over S-1 support is very weak and can be neglected with respect to its adsorption on Au. Moreover, the poor S/N ratios of the FTIR spectra is ascribed to the ultralow Au loading. In this regard, we have employed synchrotron-based FTIR (SR-FTIR) microspectroscopy with high brightness, and therefore small spot size and faster acquisition of high-quality spectral imaging data from a synchrotron light source. Accordingly, we have revised the relevant description as follows:

“Meanwhile, we have conducted the in-situ SR-FTIR measurement of C₃H₆ desorption, in which the band ranging from 2935 to 3015 cm⁻¹ was compared in Fig. 5c, where the influences of C₃H₆ adsorption on S-1 support could be excluded as shown in Supplementary Fig. S35.”

“In this regard, we conducted in-situ synchrotron-based FTIR (SR-FTIR) measurements with a

smaller spot size and faster acquisition capability, enabling us to obtain high-quality spectral imaging data from a synchrotron light source.”

Figure S35. In-situ DRIFTS measurement of C₃H₆ adsorption over S-1 at 40 °C.

5. How were the C₃H₆ conversions measured, directly by the C₃H₆ concentration changes or indirectly by calculations from the products? If the latter, errors could be quite large on the kinetic data.

Response:

Thank you for your thoughtful reminder regarding this matter. In this study, we determined C₃H₆ conversions indirectly through calculations based on the product analysis rather than tracking changes in C₃H₆ concentration. This approach is rooted in the findings of a prior study conducted by Delgass et al. (*J. Catal.*, **2018**, 365, 105–114), which highlighted that propylene conversions measured using the propylene peak were typically less than 5%, closely approaching the precision limit of the instrument. As a result, reported C₃H₆ conversions have commonly been calculated by dividing the moles of carbon-containing products detected by the moles of C₃H₆ fed (*J. Catal.*, **2012**, 287, 178–189; *ACS Catal.*, **2023**, 13, 2069–2085; *ACS Catal.*, **2017**, 7, 2668–2675; *Angew. Chem.*, **2021**, 133, 18333–18341; *Catal. Sci. Technol.*, **2018**, 8, 3052–3059). Moreover, CO₂ can be well-analyzed in this work, and the carbon balance in all experiments was found to be better than 95%. To make them clear, we have added the relevant descriptions into the revised version as follows:

“The reported C₃H₆ conversions were determined based on the moles of carbon-containing products, as the measurements using the propylene peak often yielded values lower than 5%, which is in close proximity to the instrument's accuracy threshold. The C₃H₆ conversion, PO selectivity, PO formation rate, H₂ efficiency, and carbon balance were calculated as follows:

C₃H₆ conversion = moles of (C₃-oxygenates + 2/3 ethanal + 1/3 CO₂) / moles of propylene in the feed.

PO selectivity = moles of PO / moles of (C₃-oxygenates + 2/3 ethanal + 1/3 CO₂).

PO formation rate = (C₃H₆ mole flow rate × C₃H₆ conversion × PO selectivity) / (moles of Au × reaction time)

H₂ efficiency = moles of PO / moles of H₂ converted.

Carbon balance = moles of [2 × ethanal + CO₂ + 3 × (C₃-oxygenates + propylene)] / mole of 3 × propylene in the feed.

It should be noted that the CO₂ can be well-analyzed in this work, and the carbon balance in all experiments was found to be better than 95%.”

Reviewer #2 (Remarks to the Author):

Summary: Wang et al. performed a comprehensive proof-of-concept study on the catalytic performance of silicalite-supported Au single atoms (SAs), nanoparticles (NPs), and SA/NP mixes for propylene epoxidation. It was found that at intermediate loadings of Au (0.014 wt%) resulted in dramatically improved PO formation rates. (S)TEM imaging suggests that these highly active samples contain a mixture of Au NPs and SAs. The authors hypothesize that Au SAs have favorable reaction kinetics at low C₃H₆ coverages, but become poisoned at high coverages. However, the introduction of trace Au NPs provided the “antidote” to this poisoning by providing sites for hydroperoxyl radicals to form. This hypothesis is supported by a combination of DFT and MD simulations, as well as catalyst mixing tests. The authors also provide mechanistic insights and reaction profiles supported by computation. Finally, the authors provide a descriptor to determine optimal balance between NP and SA sites based on the difference between C₃H₆ and H₂ reaction order.

General Comments:

I recommend that the scope, impact, and quality of this paper are appropriate for Nature Communications, though there are some concerns which should be addressed before publishing. While the title and abstract are written somewhat confusingly, the rest of the manuscript is easy to follow. The authors have provided a strong combination of experimental and computational studies which support their hypothesis, and this work present a new avenue for studying synergistic effects between metal NPs and SAs.

Specific Major Questions/Concerns:

1. The use of the word “antidote” in the title and abstract is somewhat confusing, particularly because poisoning isn't mentioned in the title. It's longer, but something along these lines might be easier to understand: “Gold nanoparticles provide the antidote for single atom catalyst poisoning during sustainable propylene epoxidation in the absence of titanium” or “Synergistic effects between Au single atoms and nanoparticles lead to enhanced sustainable propylene epoxidation in the absence of titanium”

Response:

Thank you for this kind suggestion. Accordingly, we have revised the title as follows:

“The Antidote of Nanoparticles for Poisoned Gold Single-Atom Catalyst toward Sustainable

Propylene Epoxidation”

2. The analytical methodology used for particle size quantification should be made more clear. How many samples/particles/images were used to determine average particle size? Can you also clarify how uniform the size distribution of Au NPs was, or provide a particle size histogram? Providing these details as well as additional (S)TEM images would strengthen claims made about the structure of the catalysts.

Response:

We are sorry for missing these important details of particle size quantification. In the revised version, we have provided the relevant descriptions, particle size histogram, as well as additional (S)TEM images as follows:

“As characterized by HAADF-STEM in Supplementary Figs. S3-S10, lowering Au loadings from 0.460 wt% to 0.026 wt% can decrease Au particle size from 3.9 to 2.1 nm (Supplementary Table S1), based on the measurement of more than 150 random particles.”

Figure S3. Typical HAADF-STEM images of Au/S-1 with the loading of 0.461 wt%.

Figure S4. Typical HAADF-STEM images of Au/S-1 with the loading of 0.230 wt%.

Figure S5. Typical HAADF-STEM images of Au/S-1 with the loading of 0.149 wt%.

Figure S6. Typical HAADF-STEM images of Au/S-1 with the loading of 0.070 wt%.

Figure S7. Typical HAADF-STEM images of Au/S-1 with the loading of 0.051 wt%.

Figure S8. Typical HAADF-STEM image of Au/S-1 with the loading of 0.026 wt%.

Specific Minor Questions/Comments:

1. It would be interesting if the authors provided insights on how an optimal ratio of Au SAs and NPs could be achieved at higher metal weight loadings

Response:

Thank you for this kind suggestion. Actually, in our previous work (*ACS Catal.*, **2021**, 11, 4146–4156), we have reported the atomic-level precise synthesis of Pt/graphene subnanocatalysts (from single atom and dimer to cluster) by atomic layer deposition, achieved by a high-temperature pulsed ozone strategy to controllably pre-create abundant in-plane epoxy groups on graphene as anchoring sites. This innovative method has the potential to be extended to our current study, facilitating the preparation of catalysts with an optimal ratio of Au single atoms and nanoparticles even under a high Au loading. By fine-tuning the microstructure and the population of anchoring sites, we can achieve the desired balance between Au SAs and NPs. In light of this, we have included an expanded discussion on this topic in the revised version as follows:

“In order to produce PO, the reactant surface coverages can be fine-tuned by the intimacy and composition of Au SAs and Au NPs. Hence, from the perspective of mesokinetics,¹⁴ $n_{C_3H_6-n_{H_2}}$ is identified as the catalytic descriptor for quantitatively correlating the microscopic properties of active sites, including the intimacy and composition of Au SAs and Au NPs, with the macroscopic catalytic performance, which can predict catalytic function and screen catalysts. Accordingly, the mortar-mixing catalyst ($Au_n:Au_1=1:7$) is suggested to achieve the coverage-matching with moderate $n_{C_3H_6-n_{H_2}}$ in Figs. 6a-6c, thus contributing to a rate-matching scenario

with the highest catalytic performance.”

“It is worth noting that these catalysts still encounter certain issues, such as a prolonged induction period and inadequate conversion at low Au loadings. These challenges can potentially be mitigated by selecting calcined S-1 as the catalyst support for achieving precise atomic-level synthesis of Au/S-1 subnanocatalysts with a higher Au loading.”

2. While the on-stream stability looks generally good, can the authors comment on the loss in activity over 100s of hours in Fig 6d? Do they believe it's due to poisoning, or perhaps catalyst sintering?

Response:

Thank you for this kind suggestion. Accordingly, we have carried out SEM, HAADF-STEM, XPS, and TGA measurements for a comparison between the fresh and spent catalysts, and the results are shown in Figs. S42-S45. It can be seen that both the agglomeration of Au nanoparticles and carbonaceous deposits contribute to the loss of catalytic activity. As a result, we have added more discussion in the revised version as follows:

“After the long-term testing, the mortar-mixing catalyst exhibited a gradual decrease in reaction rate, which could be ascribed to the slight increase in Au particle size by particles sintering (Supplementary Figs. S42 and S43) and the poisoning of Au active site by carbonaceous deposits (Supplementary Figs. S44 and S45).”

Figure S42. Typical SEM images of the fresh mortar-mixing catalyst (a), and spent mortar-mixing catalyst (b).

Figure S43. Typical HAADF-STEM images of the fresh mortar-mixing catalyst (a), and spent mortar-mixing catalyst (b).

Figure S44. XPS Au 4f spectra of the fresh and spent mortar-mixing catalyst. The shift of Au binding energy and decrease in signal for the spent catalyst indicate the poisoning by carbonaceous deposits.

Figure S45. TGA analysis of the spent mortar-mixing catalyst.

3. Are the normalized PA formation rates, such as those in Figure 5a, calculated based on total Au content or Au surface sites for samples with Au NPs (i.e. is decreased dispersion accounted for)?

Response:

Thank you for this good question, and the referee is right that the decreased dispersion could account for the volcano curves in Figure 5. Actually, the PO formation rates in Figure 5 are calculated based on total Au content rather than Au surface sites. To study the influences of Au dispersion, we have calculated the PO formation rate based on Au surface atoms and then plotted them as shown below. It can be seen that the PO formation rates based on Au surface sites exhibit almost the same trend as those based on Au total sites in Figure 5, which would help us exclude the decreased dispersion as the main cause for Figure 5.

Figure. PO formation rate based on Au surface sites for mixing Au₁ and Au_n in different intimacies (a) and compositions (b).

Figure 5. Effects of intimacy and composition of Au SAs and Au NPs.

a, b, PO formation rate and selectivity for mixing Au₁ and Au_n in different intimacies (a) and compositions (b).

Reviewer #3 (Remarks to the Author):

Overall this is a good and thoroughly thought paper. It provides some very good ideas in the behaviour of Single Atom Catalysts in regard to a technically (in terms of reaction mechanism) challenging and commercially important reaction. I believe the paper provides various new insights in the behaviour of Au atoms and clusters. I think the paper is suitable for publication in Nature Communication. However, the authors should take into account the following comments (especially comments 6).

1. I think the introduction (first paragraph) where the overall behaviour and characteristics of Single Atom Catalysts is described requires more of a recent review type of reference like <https://doi.org/10.1016/j.pecs.2023.101074> or <https://doi.org/10.1016/j.pecs.2023.101074>.

Response:

We are sorry for missing these important literatures. In the revised version, we have added them as follows:

“Single-atom catalysts (SACs) capture significant interest in catalysis by virtue of their maximum atom utilization, tunable electronic properties, and special size quantum effects, which afford superior catalytic performances across a wide range of catalytic reactions.¹⁻⁴”

3. Islam, M. J. et al. PdCu single atom alloys supported on alumina for the selective hydrogenation of furfural. *Appl. Catal. B: Environ.* **299**, 120652 (2021).

4. Loy, A. C. M. et al. Elucidation of single atom catalysts for energy and sustainable chemical production: Synthesis, characterization and frontier science. *Prog. Energ. Combust.* **96**, 101074 (2023).

2. Figure S1 is a nice and useful figure showing the energy profiles of the two pathways under consideration. However, the small images the authors have inserted are hard to see (at least with the image resolution provided in the reviewer document I have received). Please make sure that the resolution of the images is improved. (The same applies for Figure S2).

Response:

We are very sorry for the low resolution of Figures S1 and S2. In the revised version, we have improved their resolutions as follows:

Figure S1. Calculated energy profile and the corresponding structural configurations for PO formation on Au(111) surface, which involves the attack of C₃H₆* by OOH* to generate PO* and OH*.

Figure S2. Calculated energy profile and the corresponding structural configurations for acrolein formation on Au(111) surface, which involves the attack of C_3H_6^* by OOH^* to generate $\text{C}_3\text{H}_4\text{O}^*$, H^* , and OH^* .

3. In connection with the previous comment, the two mechanisms are very important for the manuscript. The authors may wish to consider to include them in the form of simple chemical equations in the main manuscript. (I understand that this may not be possible, this is mostly a suggestion).

Response:

Thank you for this kind suggestion. As the referee suggested, the mechanisms for these two reactions are quite important yet complex. On the one hand, the overall reaction equation for PO formation could be written as $C_3H_6+H_2+O_2\rightarrow C_3H_6O+H_2O$ based on previous study (ACS Catal., 2022, 12, 10147–10160). Accordingly, we have added the relevant description into the revised version as follows:

“The direct epoxidation of propylene with H_2/O_2 to value-added propylene oxide (PO) using bifunctional Au-Ti catalysts ($C_3H_6+H_2+O_2\rightarrow C_3H_6O+H_2O$) is considered a dream reaction for PO production.^{17,18}”

On the other hand, there is still no consensus on the chemical reaction equation for acrolein formation based on the current literatures. To make the elementary steps in terms of the mechanisms clearer, we have revised the captions of Figures S1 and S2 as follows:

“Figure S1. Calculated energy profile and the corresponding structural configurations for PO formation on Au(111) surface, which involves the attack of $C_3H_6^$ by OOH^* to generate PO^* and OH^* .”*

“Figure S2. Calculated energy profile and the corresponding structural configurations for acrolein formation on Au(111) surface, which involves the attack of $C_3H_6^$ by OOH^* to generate $C_3H_4O^*$, H^* , and OH^* .”*

4. Figure 1 caption, I think it would help a lot if the authors could add the gold loading in parenthesis next to Au1, Au1&n and Aun.

Response:

According to your suggestion, we have added the gold loadings into the caption of Figure 1 as follows:

Fig. 1 Structural characterization of Au₁, Au_{1&n}, and Au_n.

a-c, HAADF-STEM images in low magnification of Au₁ (**a**), Au_{1&n} (**b**), and Au_n (**c**) catalyst. **d-f**, Aberration-corrected HAADF-STEM images in high magnification of Au₁ (**d**), Au_{1&n} (**e**), and Au_n (**f**) catalyst. **g**, XPS Au 4f spectra of Au/S-1 catalysts with different loadings. **h, i**, In-situ SR-FTIR spectra of C₃H₆ adsorption (**h**) and that in the range of 2800-3100 cm⁻¹ (**i**) of the Au₁ and Au_n catalyst at 40 °C. The Au loadings for the Au₁, Au_{1&n}, and Au_n catalysts are 0.004, 0.014, and 0.026 wt%, respectively.

5. In regard to the XPS, I agree that the Au 4f shift is most likely related to final state effects, however the authors should add a relevant reference (there are plenty in the literature, so I am not suggesting a specific one).

Response:

Thanks for this kind suggestion. Accordingly, we have added the relevant reference in the revised version as follows:

“The XPS spectra in Fig. 1g exhibit a continuous increase of Au 4f binding energy by decreasing Au size, most likely attributed to the final state effect resulting from the size-dependent electrostatic interaction between ionized cluster and escaping photoelectron.³⁶”

36. Peters, S., Peredkov, S., Neeb, M., Eberhardt, W. & Al-Hada, M. Size-dependent XPS spectra of small supported Au-clusters. *Surf. Sci.* **608**, 129–134 (2013).

6. Looking overall at the manuscript, I am not sure how the authors prove that they have obtained a Single Atom Catalyst (or at least this is not well explained). From HAADF-STEM it is not clear that we are dealing with single atoms. Possibly NEXAFs could tell more (from the point of view that from the EXAFS equation they can obtain the coordination number). How are they sure that they are not dealing with Au dimers or trimers. Very likely the low loading material (Au₁) contains all the above (tiny clusters of Au and possibly single atoms too). I think this has to become very clear in the manuscript.

Response:

Thank you for kindly reminding us on this issue. Actually, as the referee suggested, we have tried to calculate the coordination number based on the analysis of the EXAFS regions in XAS spectra. Unfortunately, the XAS measurement failed to provide sufficient resolution due to the ultralow Au loadings. In this regard, we have conducted the in-situ synchrotron-based FTIR (SR-FTIR) measurement of C₃H₆ adsorption. Fig. 1i reveals a noticeable red shift of 2 cm⁻¹ for the Au_n catalyst with respect to the Au₁ catalyst. This shift is most likely attributed to the higher electron density of Au NPs compared to Au SAs. Moreover, according to the referee's suggestion, we have carried out more HAADF-STEM measurements as shown in Figure S10. It can be seen that Au₁ only exhibits Au single atoms rather than Au dimers or trimers in different selected areas. In this regard, it is reasonable to deduce that Au mainly exist as single atom over the Au₁ catalyst surface, and the relevant description has been added as follows:

“Unfortunately, XAS failed to gain more insights into their electronic structures and coordination environments due to the low Au loadings.”

“Hence, aberration-corrected HAADF-STEM was employed to characterize the catalysts with the loading of 0.004, 0.014, and 0.026 wt%, which consist of isolated Au SAs (labeled by the red circle in Fig. 1d and Supplementary Fig. S10), a mixture of Au SAs and NPs (Fig. 1e), and Au NPs (labeled by the blue square in Fig. 1f), respectively. In this regard, the three catalysts were denoted as Au₁, Au_{1&n}, and Au_n, in which the absence of Au NPs for Au₁ catalyst is consistent with its ultralow loading (0.004 wt%).”

Figure S10. Typical HAADF-STEM images of Au/S-1 with the loading of 0.004 wt%.

7. I believe that the manuscript would benefit with some more discussion on the post reaction characterization of the materials used.

Response:

Thank you for this kind suggestion. Accordingly, we have carried out SEM, HAADF-STEM, XPS, and TGA measurements for a comparison between the fresh and spent catalysts, and the results are shown in Figs. S42-S45. It can be seen that both the agglomeration of Au nanoparticles and carbonaceous deposits contribute to the loss of catalytic activity. As a result, we have added more discussion in the revised version as follows:

Thank you for your thoughtful suggestion. We conducted SEM, HAADF-STEM, XPS, and TGA measurements to compare the fresh and spent catalysts. The results are presented in Figs.

S42-S45. Our observations indicate that both the agglomeration of Au nanoparticles and the presence of carbonaceous deposits contribute to the decline in catalytic activity. Consequently, we have added more discussion in the revised version as follows:

“After the long-term testing, the mortar-mixing catalyst exhibited a gradual decrease in reaction rate, which could be ascribed to the slight increase in Au particle size by particles sintering (Supplementary Figs. S42 and S43) and the poisoning of Au active site by carbonaceous deposits (Supplementary Figs. S44 and S45).”

Figure S42. Typical SEM images of the fresh mortar-mixing catalyst (a), and spent mortar-mixing catalyst (b).

Figure S43. Typical HAADF-STEM images of the fresh mortar-mixing catalyst (a), and spent mortar-mixing catalyst (b).

Figure S44. XPS Au 4f spectra of the fresh and spent mortar-mixing catalyst. The shift of Au binding energy and decrease in signal for the spent catalyst indicate the poisoning by carbonaceous deposits.

Figure S45. TGA analysis of the spent mortar-mixing catalyst.

Reviewer #4 (Remarks to the Author):

The manuscript by Wang, Sang, Liu et al. presents interesting and novel results based on a sophisticated methodology in the search for a new catalyst for propylene epoxidation that avoids the problems of the single Au atom catalyst supported on silicalite-1 and Ti-silicalite-1. The catalyst suggested by the authors is based on the mix of Au NPs and Au SAC that avoids the poisoning of Au SAC by generating hydroperoxyl on the Au NPs. The experimental results suggest the increase in the catalytic activity of the mixed catalyst while the characterization of the material confirms the existence of SAC and NPs by means of a method that mixes different weight loads. The theoretical counterpart gives solid evidence that contributes to the understanding of the mechanisms that allow the mixed catalyst to improve the catalytic activity of the system. The results are very significant in the field, and I recommend widely its publication in Nature Communications. However, there are some questions that could help to clarify the synergetic effect of the Au NPs and Au SA in the mixed catalyst. The issues that could help to improve the quality of the manuscript are the following:

1) The diffusion barrier of Au atoms on silicalite-1 could be calculated by means of DFT calculations to know the capability of Au atoms to sinter and create larger NPs and reduce the catalytic activity of the mixed catalyst.

Response:

Following the referee's suggestion, we have employed DFT calculations to determine the diffusion barrier of Au atoms on silicalite-1. The findings, detailed below, reveal a significant diffusion barrier of 2.66 eV. This indicates a high stability of Au single atom against sintering on the silicalite-1 surface. Further details are addressed in our response to the second issue.

Figure. Potential energy profiles for diffusion of Au single atom over silicalite-1.

2) The MD without the molecules could also show the possibility of the sintering process.

Response:

Thank you for the kind suggestion. We have conducted molecular dynamics (MD) simulations on the Au₁ catalyst without molecules to examine the ability of Au atoms to sinter and form larger nanoparticles (NPs). The results are presented in Figure S41. It is evident that the majority of Au single atoms remain stable on the S-1 surface, with only a few instances of agglomeration. This observation can be attributed to the catalyst's exceptional stability. Consequently, we have incorporated a relevant discussion in the revised version as follows:

“The exceptional stability observed can likely be attributed to the robust interactions between Au single atoms and S-1 support, as depicted in Supplementary Fig. S41, which effectively hinder site agglomeration during the reaction.”

Figure S41. The simulation of the sintering process for the Au₁ catalyst.

3) The contribution of the vdW forces in the molecule-Au interaction is very important. Could you separate that contribution to know how big it is? Could you do a single-point or relaxation calculation with a more accurate method to see that the results are consistent and do not depend on the vdW forces approximation?

Response:

Thank you for kindly reminding us on this issue. As the referee suggested, the contribution of the vdW forces in the molecule-Au interaction is very important, and we employed the DFT-

D3 correction scheme to assess the weak van der Waals interactions between the adsorbed molecules and the catalyst surface. However, to specifically assess the contribution of vdW forces, we conducted a single-point calculation without the DFT-D3 correction, and the results are shown below. It can be observed that there is a change in the adsorption energy, particularly for C_3H_6 on the Au(111) surface, when the DFT-D3 correction is excluded. Despite this change, the overall trend remains consistent with our previous findings. Notably, H_2 exhibits very weak binding with both the Au single atom, Au_{13} cluster, and Au(111) surface. On the other hand, both O_2 and C_3H_6 display significantly higher adsorption energies on the Au single atom and Au_{13} cluster compared to their adsorption on the Au(111) surface.

Figure. The comparison of adsorption energies (ΔE_{ads}) of O_2 , H_2 , and C_3H_6 over Au single atom, Au_{13} cluster, and Au(111) surface (a) with and (b) without DFT-D3 correction.

4) The temperature in the MD simulation is not explained in detail. Is the temperature constant or is there a temperature ramp? What is the size of the step? Why that REAXFF potential is employed? Could you include the comparison of adsorption energies between REAXFF potential and DFT?

Response:

We are sorry for missing these important details. Actually, we decided to utilize the ReaxFF potential due to its extensive usage as a force field model in depicting chemical reactions within intricate molecular systems. Its favorable adjustability and flexibility make it well-suited for simulating various chemical reactions and material properties. By employing ReaxFF in our

research, we are able to effectively simulate the interactions between gas molecules and the gold surface, including potential chemical reactions. This is essential for comprehending the adsorption behavior and gas interactions involved. Moreover, ReaxFF offers a quantitative portrayal of the interactions between gas molecules and the gold surface, enabling us to gain deeper insights into the underlying mechanisms of this process. In this regard, we have added the relevant description in the revised version as follows:

“The system temperature was initially at 0 K and then increased rapidly at a rate of 20 K/ps until it reached 298.15 K. Following this, the system equilibrated at 298.15 K for a duration of 50 ps. In each simulation run, a Berendsen thermostat with a damping constant of 100 fs was utilized to control the temperature. During the initial equilibrium relaxation stage, a time step of 0.25 fs was employed. Once the equilibrium was achieved, the temperature was raised from 298.15 K to the desired preset temperature of 473.15 K. This temperature increase was performed gradually with a heating rate of 35.6 K/ps. Subsequently, the system remained at this elevated temperature for a simulation period of 2 ns.”

Figure. Simulation profile of temperature versus time.

Furthermore, we sincerely appreciate your suggestion regarding the comparison of adsorption energies between ReaxFF potential and DFT. In response, we have endeavored to calculate adsorption energies based on the ReaxFF potential. Specifically, the electronegativity equalization method (EEM) is used to determine the charges on the atoms, which in turn determine the Coulomb energy contribution to the total energy. In this context, when dealing with an isolated molecule, the redistribution of charge is confined to the atoms within the molecule. Similarly, in the

case of an isolated slab, charge redistribution is restricted to the atoms within the slab. However, within the combined system of slab and molecule, there is greater flexibility in charge redistribution, offering the potential for energy reduction and achieving more stable adsorption. Such challenge bears a resemblance to the basis set superposition error, and thus it is inadequate to calculate the adsorption energy using the same approach as DFT calculations ($\Delta E_{\text{ads}} = E_{\text{slab+mol}} - E_{\text{slab}} - E_{\text{mol}}$). Instead, the in-cell approach ($\Delta E_{\text{ads}} = E_{\text{slab+mol}} - E_{\text{slab+mol widely separated}}$) is employed for the calculation. Here, $E_{\text{slab+mol widely separated}}$ represents the energy for a system where the slab and molecule are placed far apart in the same cell (“in-cell”). The calculated adsorption energies are presented in Figure S47 and further compared with those obtained through DFT calculations in Figure S48. Notably, both methods demonstrate consistent trends in the adsorption capabilities of various species on the Au (111) surface and Au single atom. Based on these findings, we have incorporated relevant discussions in the revised version as outlined below:

“The reactants adsorption based on ReaxFF simulations was also investigated in Figure S47, and a further comparison of the adsorption energy between ReaxFF simulations and DFT calculations is made in Figure S48. It can be seen that both methods demonstrate consistent trends in the adsorption capabilities of various species on the Au (111) surface and Au single atom.”

Figure S47. The adsorption configurations and corresponding energies (ΔE_{ads}) of C_3H_6 , O_2 , and H_2 on Au(111) and Au single atom.

Figure S48. The comparison of the adsorption energy between DFT calculations (a) and ReaxFF simulations (b).

5) The author talks about the ionized character of Au SA. The theoretical calculation of the charge distribution (NBO, Bader, or other method) or the shift 4f states could corroborate this result.

Response:

Thank you for the kind suggestion. Following your suggestion, we have performed Bader charge analysis and projected density of state (PDOS) analysis on the Au atom supported on S-1. The Bader charge analysis reveals a negligible electron transfer (only 0.01 e) from the Au atom to S-1, which seems opposite to the XPS result of much electron transfer from Au atom to S-1. However, it is crucial to note that the Bader charge analysis considers the total number of electrons, not just the valence electrons (in this case, the 5d electron for Au). To address this discrepancy, we specifically examined the Au 5d band using PDOS analysis. Through integration up to the Fermi level, we determined that the number of 5d electrons is 8.82 e. This

suggests almost one d electron transfer from Au atom to S-1, which is quite consistent with the XPS results. As a result, we have added the relevant description into the revised version as follows:

“The charge population for Au atom, obtained by integrating the projected density of state (PDOS) up to the Fermi level as shown in Supplementary Figure S18, is determined to be 8.82 e . This implies an almost complete transfer of one d electron from the Au atom to S-1, which is consistent with the observed binding energy shift in XPS spectra.”

Figure S18. PDOS analysis of Au 5d states.

REVIEWER COMMENTS

Reviewer #1 (Remarks to the Author):

In the revised submission, the authors have replied to my previous comments and revised the manuscript accordingly. However, the following issues must be clarified before the manuscript can be accepted.

1) In their replies to my previous comment 2, the authors claimed that the long induction period can be related to the decomposition of the template absorbed on the external surfaces of the uncalcined titanium silicates. However, in the present manuscript, S-1, instead of TS-1, was used as the support. They need to identify if the Au SCs remained stable during the long induction period. I am sure that the authors are aware of the strong argument whether the very fine Au clusters within the channels of TS-1 catalysts are responsible for the propylene epoxidation with H₂ and O₂.

2. In their replies to my previous comment 3, the authors claimed that this negative shift can be attributed to the transfer of electrons from the S-1 support to the gold particles, resulting in the formation of anionic gold species. But S-1 is pure SiO₂, and generally charge transfer barely occurs from SiO₂ to Au. The authors need to provide additional evidence. Additionally, in the Fig. S44, the Au 4f binding energy of used catalysts came back to 83.9 eV, the normal value. Such a shift can not be explained by agglomeration of Au species during the catalytic reaction.

Since the structures of Au are the core of the manuscript, the authors must address these issues to prove that their argument holds.

Reviewer #3 (Remarks to the Author):

The authors have revised thoroughly the manuscript and addressed all of my comments. I believe they have also addressed very well the comments of other referees. I think the manuscript can be published without further revision.

Reviewer #4 (Remarks to the Author):

The answer to the questions fulfills all the points that I suggest and introduces them in the manuscript in a good way. I recommend publishing the paper in its current form.

Many thanks for the valuable comments and suggestions from the reviewer. We have revised our paper by fully taking into account all the comments and suggestions.

Reviewer #1 (Remarks to the Author):

In the revised submission, the authors have replied to my previous comments and revised the manuscript accordingly. However, the following issues must be clarified before the manuscript can be accepted.

1. In their replies to my previous comment 2, the authors claimed that the long induction period can be related to the decomposition of the template absorbed on the external surfaces of the uncalcined titanium silicates. However, in the present manuscript, S-1, instead of TS-1, was used as the support. They need to identify if the Au SCs remained stable during the long induction period. I am sure that the authors are aware of the strong argument whether the very fine Au clusters within the channels of TS-1 catalysts are responsible for the propylene epoxidation with H₂ and O₂.

Response:

Thank you for the considerate suggestion. In this work, we used the uncalcined S-1 with blocked micropores as the support to prepare Au catalysts. As a result, the Au species are deposited on the external surfaces of the support, avoiding the deactivation caused by PO adsorption and micropore blocking. However, these Au species can migrate into the micropores with the decomposition of template during the induction period. Moreover, as noted by the reviewer, there is also an ongoing debate regarding the role of ultrafine gold (Au) clusters within the channels of TS-1 catalysts in the propylene epoxidation process with H₂ and O₂. Hence, following the reviewer's suggestion, we conducted tests to assess the stability of Au single atoms by characterizing the Au₁ catalyst after the induction period. The results, depicted in the figure below, show that the Au single atoms remain stable across the catalyst surface.

Figure. Typical HAADF-STEM image of the Au₁ catalyst after the induction period.

Additionally, we employed Density Functional Theory (DFT) calculations to determine the diffusion barrier of Au single atoms on silicalite-1. The detailed findings reveal a significant diffusion barrier of 2.66 eV, indicating a high stability of Au single atoms against sintering on the silicalite-1 surface.

Figure. Potential energy profiles for the diffusion of Au single atom over silicalite-1.

It is noteworthy that the DFT calculations are predictive for Au single atoms in low density. To complement this, we conducted Molecular Dynamics (MD) simulations to explore the possibility of sintering processes in high density. The results, presented below, show that the majority of Au single atoms remain stable on the S-1 surface, with only a few instances of agglomeration. This observation can be attributed to the catalyst's robust stability. Consequently, we have incorporated a relevant discussion in the revised version as follows:

“The observed high stability is likely due to the robust interactions between Au single atoms

and the S-1 support, as depicted in Supplementary Fig. S41, effectively hindering site agglomeration during the reaction.”

Figure S41. The simulation of the sintering process for the Au₁ catalyst.

2. In their replies to my previous comment 3, the authors claimed that this negative shift can be attributed to the transfer of electrons from the S-1 support to the gold particles, resulting in the formation of anionic gold species. But S-1 is pure SiO₂, and generally charge transfer barely occurs from SiO₂ to Au. The authors need to provide additional evidence. Additionally, in the Fig. S44, the Au 4f binding energy of used catalysts came back to 83.9 eV, the normal value. Such a shift can not be explained by agglomeration of Au species during the catalytic reaction. Since the structures of Au are the core of the manuscript, the authors must address these issues to prove that their argument holds.

Response:

Thank you for your insightful question. Following your suggestion, we conducted Bader charge analysis on the Au atom supported on S-1. The Bader charge analysis revealed a minimal electron transfer (only 0.01 e) from the Au atom to S-1, which seems contrary to the XPS result. However, it is essential to note that the templates, including TPAOH, absorbed on the uncalcined S-1 surfaces could serve as electron-donating species. This phenomenon has been observed to facilitate electron transfer to supported metals or metal oxides in previous studies (e.g., J. Environ. Sci., 2021, 107, 87–97; Surf. Interfaces, 2021, 22, 100897). In this regard, these templates can donate abundant electron to supported Au species and lower their binding energy to 83.42 eV compared to bulk Au. Moreover, we fully agree with the reviewer that the Au binding energy shift from 83.42 eV to 83.92 eV can not be explained by agglomeration of

Au species during the catalytic reaction. Considering the decomposition of the aforementioned templates after the induction period, the electron donation by templates to Au species will be hindered. This provides a plausible explanation for the remarkably positive binding energy shift observed in Fig. S45. Accordingly, we have added the relevant discussion in the revised version as follows:

“Moreover, the significant shift in Au 4f binding energy, as shown in Supplementary Fig. S45, is very likely to have originated from the decomposition of tetrapropylammonium hydroxide (TPAOH), which serves as the structure-directing agent for S-1.”

Figure S45. XPS Au 4f spectra of the fresh and spent mortar-mixing catalyst. The decrease in signal for the spent catalyst indicate the poisoning by carbonaceous deposits.

REVIEWERS' COMMENTS

Reviewer #1 (Remarks to the Author):

The revised manuscript can be accepted.

Reviewer #1 (Remarks to the Author):

The revised manuscript can be accepted.

Response:

We appreciate the reviewer for the positive comments and the previous valuable suggestions.